

# Exploring the complexities associated with full-scale wind plant wake mitigation control experiments

## James B. Duncan Jr.[1], Brian D. Hirth[1], John L. Schroeder[2]

[1]National Wind Institute, Texas Tech University, Lubbock, 79409, USA
[2]Department of Geosciences, Texas Tech University, Lubbock, 79409, USA

*Correspondence to*: James B. Duncan Jr. (james.b.duncan@ttu.edu)

**Abstract.** Recent research promotes implementing next-generation wind plant control methods to mitigate turbine-to-turbine wake effects. Numerical simulation and wind tunnel experiments have previously demonstrated the potential benefit of wind plant control for wind plant optimization, but full-scale validation of the wake-mitigating control strategies remains limited. As part of this study, the yaw and blade pitch of a utility-scale wind turbine were strategically modified for a limited time period to examine wind turbine wake response to first-order turbine control changes. Wind turbine wake response was measured using Texas Tech University's Ka-band Doppler radars and dual-Doppler scanning strategies. Results highlight some of the complexities associated with executing and analysing wind plant control at full-scale using brief experimental control periods. Some difficulties include (1) the ability to accurately implement the desired control changes, (2) identifying reliable data sources and methods to allow these control changes to be accurately quantified, and (3) attributing variations in wake structure to turbine control changes rather than a response to the underlying atmospheric conditions (e.g. boundary layer streak orientation, atmospheric stability). To better understand wake sensitivity to the underlying atmospheric conditions, wake evolution within the early-evening transition was also examined using a single-Doppler data collection approach. Analysis of both wake length and meandering during this period of transitioning atmospheric stability indicate the potential benefit and feasibility of wind plant control should be enhanced when the atmosphere is stable.

## 1 Introduction

During wind turbine operation, momentum is extracted from the inflow creating a waked region downstream containing less wind speed and more turbulence relative to the inflow (Manwell et al., 2009). When arranged in a dense array (i.e. a wind plant), wind turbine wakes can impact the performance (both power and loading) of downstream turbines (e.g. Barthelmie et al., 2007; Barthelmie and Jensen, 2010; Barthelmie et al., 2010; González-Longatt et al., 2012; Adaramola and Krogstad 2011, McKay et al., 2012; Schepers et al., 2012; Kim et al., 2015; El-Asha et al., 2017). This wake effect causes wind plant power production to typically be less than the performance of a single, stand-alone wind turbine scaled up to the number of turbines in the plant. Wake-related power losses can be significant and are heavily dependent on wind turbine spacing and atmospheric



conditions, such as wind speed, wind direction, and boundary layer stability. In the Horns Rev wind plant, wake-related power losses were estimated at 12.4% (Sørensen et al., 2006) and in the Lillgrund wind plant losses were estimated to be as high as 23% (Dahlberg and Thor, 2009). In addition, losses can be significantly higher for individual turbine pairs (e.g. upwards of 60 to 80% [El-Asha et al., 2017]). Therefore, despite innovations to wind turbine component design (e.g. improvements to the

blade, generator, and tower), wake-related power losses will remain a limiting factor to wind plant performance.

Wind turbine operation impacts wake structure—the more momentum extracted by the turbine, the larger the wake deficit. The controllers of individual turbines in a wind plant act autonomously, and in order to maximize power production, operate each turbine in the plant at peak capacity. However, maximizing individual turbine power extraction also increases the wake

effect. Therefore, this 'greedy' control ideology is not suitable for wind plant optimization. Wind plant control (also referred to as cooperative wind turbine control or active wake control) is a next-generation control ideology designed to mitigate the wake effect (Knudsen et al., 2015). When wind plant control is employed, some turbines in the wind plant will operate below their peak capacity to decrease the wake effect, thereby increasing the plant-wide available kinetic energy (De-Prada-Gil et al., 2015). Turbine controls are typically modified to either (1) reduce wake intensity or (2) deflect the wake away from

downstream turbine inflow regions (i.e. wake steering). The benefit of wind plant control has been previously demonstrated using numerical simulation (e.g. Johnson and Fritsch, 2012; Lee et al., 2013; Annoni et al., 2015; Park and Law, 2015; Fleming et al., 2015; Gebraad and van Wingerden, 2015; Gebraad et al., 2016; Kanev et al., 2018) and in wind tunnel experiments (e.g. Parkin et al., 2001; Corten and Schaak, 2003; Jiménez et al., 2010; Park et al., 2016; Vollmer et al., 2016; Schottler et al., 2017; Bartl et al., 2018; Fleming et al., 2018; Bastankhah and Porté-Agel, 2019). For example, research by Park and

Law (2015) indicates wind plant efficiency could be improved by 7.14 % in the Horns Rev offshore wind plant by implementing wind plant control. Despite these encouraging results, full-scale validation of wind plant control and measurement of wind turbine wake response to turbine control changes remains limited (e.g. Marathe et al., 2016; Trujillo et al., 2016; Fleming et al., 2017a; Fleming et al., 2017b; Ahmad et al., 2019; van der Hoek et al., 2019; Fleming et al., 2019).

To expand upon existing full-scale validation efforts, agreements were made with an industry partner to modify the yaw and blade pitch of a utility-scale wind turbine for a limited time period to examine the resulting variations in wake structure. Wake measurements were made using Texas Tech University's Ka-band (TTUKa) Doppler radars employing dual-Doppler (DD) scanning strategies. Results highlight some of the complexities associated with executing and analysing wind plant control strategies at full-scale using brief experimental periods.  For instance, information to accurately quantify the implementation

of a control strategy is imperative to attributing the resulting variations in wake structure to the control change itself. Given wind turbine wake structure is also sensitive to the underlying atmospheric conditions and inflow variability (e.g. Barthelmie et al., 2013; Machefaux et al., 2016; Lee and Lundquist, 2017; Subramanian and Abhari, 2018), atmospheric conditions must also be simultaneously well documented and understood.  To highlight wake sensitivity to atmospheric stability, and to



determine how atmospheric stability impacts wind plant control efforts, onshore wake evolution within the early-evening transition (EET) was also investigated using a single-Doppler (SD) data collection approach.

## 2 TTUKa radar instrumentation and deployment specifics

The TTUKa radars were used to spatially map wind plant complex flow structure and variability. These long-range scanning instruments are mobile and utilize a 0.33° half-power beamwidth, a nominal range gate spacing of 15 m, and horizontal scan speeds of 30° s$^{-1}$ to provide high-resolution spatial measurements of the atmospheric boundary layer (ABL) winds across a region (~ 100 km$^2$). To resolve the horizontal structure of the flow, the azimuth of the radar scan is varied at a fixed elevation angle. A single scan made at a fixed elevation tilt is referred to as a sector scan.  When only a single radar is available, a SD measurement strategy is used, and repetitive sector scans collect line-of-site velocities (i.e. the motion of the wind directly towards or away from the radar) with revisit times on the order of a few seconds.  Multiple radars are used to enable DD synthesis, via the collection of coordinated sector scans using multiple elevation tilts across a common area.  These sector scans can then be used to construct a three-dimensional volume of the horizontal velocity vectors (i.e. the u and v wind components) within the DD domain (Lhermitte, 1971). The DD domain is defined as the region where the two radar scan regions spatially overlap and the beam crossing angles support DD synthesis (Davies-Jones, 1979). The TTUKa radars were previously shown to be an effective tool for documenting wind plant complex flow structure and variability at relevant scales of motion using both SD (Hirth et al., 2012) and DD (Hirth and Schroeder, 2013; Hirth et al., 2015; Hirth et al., 2016) data collection approaches. The technical specifications of the TTUKa radars are further detailed in Table 1.

| Parameter | General Specification (14 December 2014 \| 12 October 2015) |
|---|---|
| Peak Transmit Power | 212.5 W |
| Transmit Frequency | 35 GHz |
| Half-power beamwidth | 0.33° |
| Azimuthal resolution | 0.352° |
| Horizontal Scan Speed | 30° s$^{-1}$ |
| Pulse width | 20 $\mu s$ |
| Range gate spacing | 15 m |
| Pulse repetition frequency | 12 kHz \| 15 kHz |
| Maximum range | 12.5 km \| 10 km |

**Table 1.** TTUKa radar(s) technical specifications for the 14 December 2014 (DD) and 12 October 2015 (SD) radar deployments. Multiple values were provided if the technical specifications between the two radar deployments varied.

### 2.1 14 December 2014 DD TTUKa radar deployment

The TTUKa radars were deployed to the north of an operational wind plant on 14 December 2014 to document changes in wind turbine wake structure due to both wind turbine yaw and blade pitch angle adaptations (Fig. 1a). Synchronized in time, the radars performed 50° sector scans across 14 elevation tilts angles (i.e. from 1.2° to 2.5° in intervals of 0.1°) to acquire a three-dimensional data volume in approximately one minute (60.4 s on average). A total of 70 DD volumes were collected



between 14:22:32 UTC and 15:31:57 UTC (hereinafter referred to as the DD analysis period). A Barnes interpolation scheme (Barnes 1964) was used to interpolate the spatially distributed radial velocity measurements from their native polar coordinate system onto a Cartesian grid. The Cartesian grid was defined by a lateral grid spacing of 10 m and was vertically defined at 10-m intervals between 30 m and 130 m (i.e. at 10-m intervals through the vertical depth of the wind turbine rotor sweep)

using a terrain-following framework. Surface elevation data (i.e. a one-arc second digital raster) from the United States Geological Survey National Elevation Dataset (http://ned.usgs.gov) was used to develop the terrain-following grid, thereby ensuring the Cartesian grid was defined at distinct constant-height intervals above-ground level (AGL). Within the DD domain (shaded region in Fig. 1a), the horizontal wind field was resolved (Fig. 1b). The DD wind maps can be interpreted as a pseudo-average of the wind conditions over the volume acquisition period, where the DD volume time stamp denotes the end of the

volume period. Although in-situ meteorological measurements were not available, the presence of boundary layer streak features in the DD wind maps suggests near-neutral ABL stability (Deardorff, 1972; Lin et al., 1996; Drobinski and Foster, 2003).

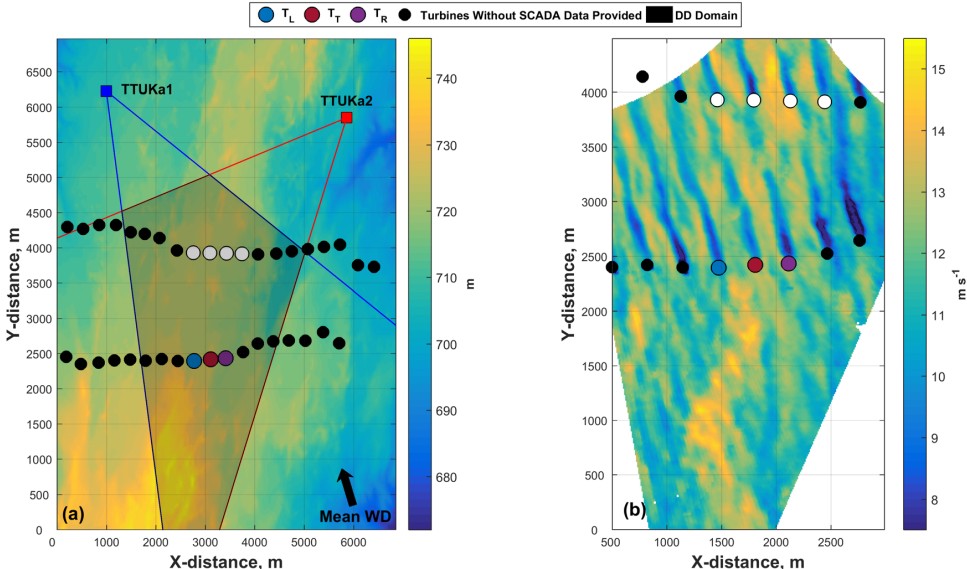

**Figure 1.** (a) Schematic of the TTUKa DD radar deployment on 14 December 2014 including the radar sectors scanned (red and blue lines), the DD domain (shaded region), the location of individual wind turbines (colored circles [black circles denote turbines whose SCADA data was not provided]), the mean wind direction (black arrow), and the underlying mean sea level elevation (m). (b) TTUKa DD hub-height wind speed (m s$^{-1}$) at 14:59:29 UTC overlaid by the wind turbine locations.

*2.2 12 October 2015 SD TTUKa radar deployment*

On 12 October 2015, a single TTUKa radar was deployed to examine wind turbine wake response to changes in ABL stability consistent with the EET prevalent in the US Great Plains (Fig. 2). Between 23:32:08 UTC and 00:52:30 UTC, the radar performed a series of 1791 SD sector scans at a one-degree elevation tilt yielding an average sector revisit time of 4.7 s.



Contained within the measurement domain was an instrumented meteorological tower providing wind, temperature, moisture, and pressure information at multiple vertical levels and a utility-scale wind turbine. The orientation of the scanned sector relative to the mean wind direction during the SD data acquisition period from the east-northeast allowed the wind field to be well-resolved using a SD data collection approach. A Barnes OA scheme was used to interpolate the polar radial velocity fields

onto a Cartesian gridded domain with a lateral grid spacing of 20 m. However, because measurements were made at a one-degree elevation tilt, the height AGL of the gridded domain increased at a rate of 17.5 m per km moving away from the radar. The measurement height of the radar sector at the location of the wind turbine was 107.8 m AGL. This measurement height was slightly above the wind turbine hub height but within the turbine wind rotor sweep; however, further proprietary turbine information (e.g. hub height, rotor diameter) cannot be disclosed.

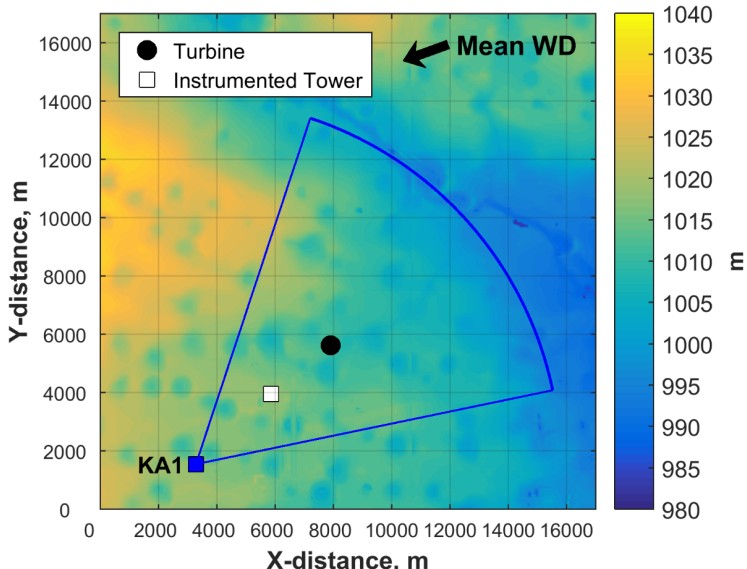

**Figure 2.** Schematic of the TTUKa SD radar deployment on 12 October 2015. Mapped is the underlying mean sea level elevation (m) overlaid by the location of TTUKa1 (blue square), the radar sector scanned (blue circle arc), the mean wind direction (black arrow), the instrumented meteorological tower (white square), and the wind turbine (black circle).

**3 Wind plant control experimental setup, controller assessment, and controller assessment challenges**

Located in the DD domain of the 14 December 2014 deployment were 20 wind turbines distributed across two turbine rows. The wind turbines were characterized by a hub height of 80 m and a rotor diameter (RD) of 101 m. Supervisory control and data acquisition (SCADA) information detailing the turbine inflow wind speed (subject to the nacelle transfer function [NTF]), turbine yaw orientation, and blade pitch angle were provided at a one-hertz sampling frequency from 14:00:00 UTC to

16:59:45 UTC for seven of the wind turbines (denoted by the non-black circles in Fig. 1). Three of the seven wind turbines were located in the leading row of the wind plant, while the remaining four were located in the trailing row. The three lead-



row wind turbines were separated by an average distance of 1512.2 m (~15 RD) from the trailing turbine row and were laterally separated from each other by an average distance of 321.1 m (~3 RD).

Experimental control changes were made to a single lead-row wind turbine, referred to as the test turbine (i.e. $T_T$) and denoted
by the red circle in Fig. 1. To examine the impact of blade pitch on wake intensity, blade pitch angle offsets of +1°, +2°, and +3° were implemented by the wind plant operator, and a +10° yaw error (positive values indicating counterclockwise rotor rotation) was employed to examine the impact of yaw error on wake deflection. Each wind turbine control change was independently implemented for a 10-min period (Table 2), meaning blade pitch and turbine yaw adaptations were not concurrently employed. Prior to analyzing wake response to wind turbine control changes, the ability of the $T_T$ controller to
accurately implement the desired control offsets was examined. The performance of the $T_T$ controller was compared to the performance of the wind turbines to both the left ($T_L$) and right ($T_R$) of the $T_T$. In Fig. 1, the $T_L$ is denoted by the blue circle and the $T_R$ is denoted by the purple circle.

| Blade Pitch Angle Offset | Time Period |
|---|---|
| +1° | 14:30:00 UTC – 14:39:59 UTC |
| +2° | 14:40:00 UTC – 14:49:59 UTC |
| +3° | 14:50:00 UTC – 14:59:59 UTC |
| Yaw Error Offset | |
| +10° | 15:22:00 UTC – 15:31:59 UTC |

**Table 2.** Wind turbine control experiment summary for the 14 December 2014 DD deployment.

*3.1 $T_T$ controller assessment*

Due to the proprietary nature of the information, wind turbine controller design was not provided by the turbine manufacturer. Not having access to wind turbine controller design is a major challenge to fully understanding wind turbine behaviour (Fleming et al., 2018), or rather, how the controller responds to variable inflow conditions when attempting to enact the desired control offsets. Therefore, the provided discussions do not detail why the turbine was able or unable to enact the desired control
changes, but rather focuses on quantifying the resulting offsets. Recommendations detailing how this analysis could be improved are also provided. These recommendations can be used to inform future full-scale validation efforts.

*3.1.1 Blade pitch controller assessment*

The $T_T$ is a pitch-regulated variable speed wind turbine. To achieve pitch control objectives, the pitch drive modifies blade
angle of attack relative to the characteristics of the inflow. Below the rated wind speed (i.e. region two), blade pitch remains fixed to maximize turbine power extraction. Alternatively, above the rated wind speed (i.e. region three), blade pitch is actively modified according to the turbine inflow wind speed to maintain the rated generator speed and stably produce the rated power of the turbine. The benefit of modifying blade pitch for wake mitigation is expected to be greatest in region two (van der Hoek et al., 2019). Consistent with region two pitch operation, the amount of momentum extracted by the wind turbine is maximized,
and therefore, wake intensity and the potential for wake-related power losses are also amplified. Furthermore, accurately





implementing blade pitch angle changes might be more feasible in region two because blade pitch remains fixed (i.e. invariable of the region two turbine inflow wind speed).

Between 14:30:00 UTC and 14:59:59 UTC, the $T_T$ was instructed to implement blade pitch angle offsets of +1°, +2°, and +3°
(experimental period durations are defined in Table 2). During the experimental periods, the $T_T$ inflow wind speeds (as defined by the nacelle anemometer) were primarily contained to region three. The $T_T$ mean inflow wind speed was 14.26 m s$^{-1}$ during the +1° experimental period, 14.32 m s$^{-1}$ during the +2° experimental period, and 13.67 m s$^{-1}$ during the +3° experimental period. Although the benefit of implementing pitch-based wake mitigation are not expected to be optimized in region three, there is still merit in examining the ability of the $T_T$ to enact the desired blade pitch angle offsets. To maintain the rated
generator speed in region three, the wind turbine follows a pitch schedule to extract the desired amount of momentum at various wind speeds. However, the pitch schedule is proprietary, and therefore, analysis of the provided SCADA information was employed to estimate it. The region three pitch schedule was constructed by fitting a linear model to the distribution of blade pitch angles as a function of the turbine inflow wind speeds. Data from all seven wind turbines were used to ensure a robust estimate of the region three pitch schedule (a total of 41,343 SCADA wind speed and blade pitch angle measurements
were used); measurements inconsistent with region three pitch operation and $T_T$ data from the experimental periods were not considered when constructing the pitch schedule.

The ability of the $T_T$ controller to accurately implement the desired control offsets was quantified by comparing blade pitch angle activity within the +1°, +2°, and +3° experimental periods to the region three pitch schedule. The residual difference
between the operational blade pitch angle (as defined by the SCADA) and the optimal value (as defined by the turbine inflow wind speed and the region three pitch schedule) was used to define the control change implemented. On average, the $T_T$ exhibited a blade pitch angle offset of -0.05° during the +1° experimental period, a blade pitch angle offset of +0.68° during the +2° experimental period, and a blade pitch angle offset of -0.03° during the +3° experimental period (Fig. 3). Due to the reactive nature of the wind turbine controller, the $T_T$ was not expected to precisely implement the desired blade pitch angle
offsets on a second-by-second basis. Even so, compared to the blade pitch behaviour of the $T_L$ and $T_R$ (Fig. 4), wherein there were no prescribed turbine control changes, there was almost no evidence to indicate the effective implementation of distinct blade pitch angle offsets by the $T_T$ controller in any of the experimental periods. Within the +1° and +3° experimental periods, the $T_T$ exhibited the smallest blade pitch angle offset of all three turbines considered, and in the +2° experimental period, the $T_T$ blade pitch angle offset was similar to that of the $T_L$. The mean blade pitch angle residual of the $T_L$, $T_T$, and $T_R$ in each
experimental period is provided in Table 3.



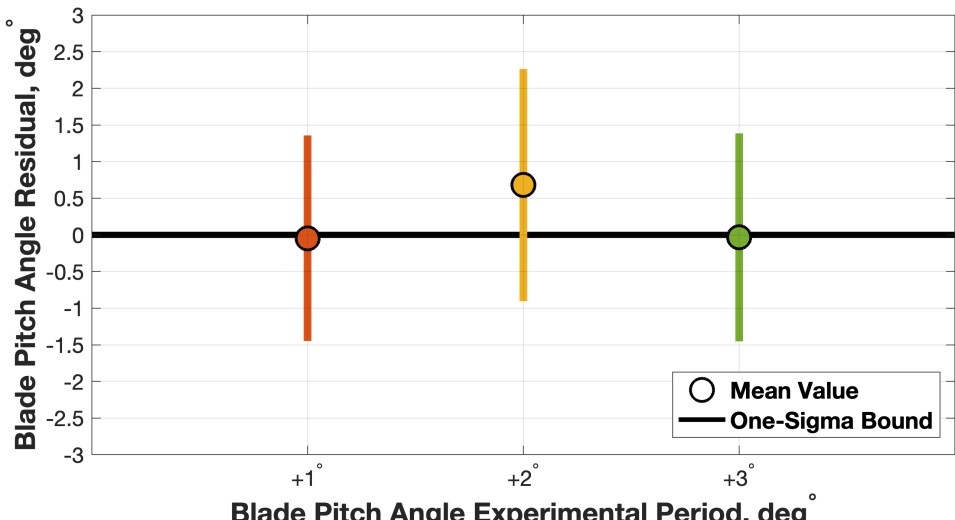

**Figure 3.** The mean and one standard deviation bound of the $T_T$ blade pitch angle residual in the +1°, +2°, and +3° experimental periods.

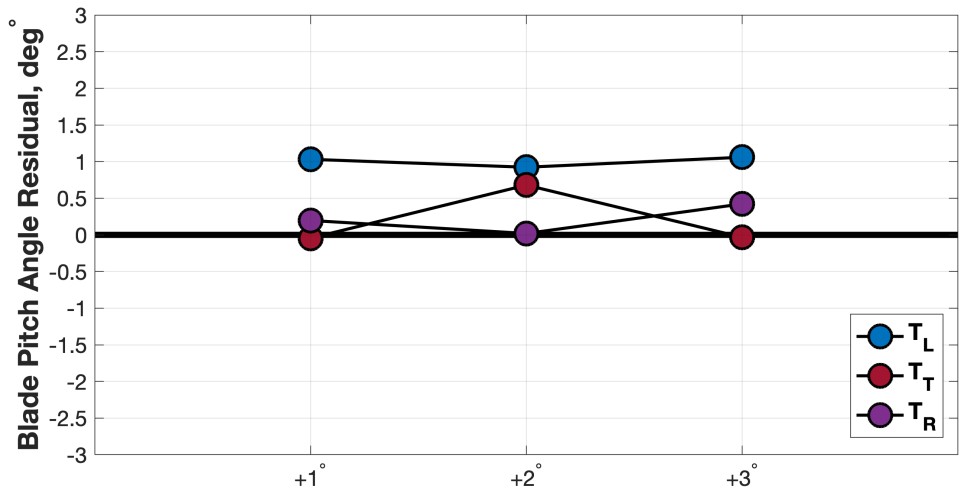

**Figure 4.** The mean $T_L$, $T_T$, and $T_R$ blade pitch angle residual in the +1°, +2°, and +3° experimental periods.

| Turbine Identifier | +1° Experimental Period | +2° Experimental Period | +3° Experimental Period | Data from all Experimental Periods |
|---|---|---|---|---|
| $T_L$ | +1.03° (51.5 %) | +0.92° (86.8 %) | +1.06° (60.2 %) | +0.99° (66.2 %) |
| $T_T$ | -0.05° (90.5 %) | +0.68° (90.2 %) | -0.03° (81.0 %) | +0.21° (87.2 %) |
| $T_R$ | +0.20° (70.7 %) | +0.02° (86.0 %) | +0.42° (60.5 %) | +0.19° (72.4 %) |

**Table 3.** The $T_L$, $T_T$, and $T_R$ mean blade pitch angle residual in the +1°, +2°, and +3° experimental periods, as well as using data from each experimental period. The values in parenthesis denote the percentage of SCADA data within each experimental period (upwards of 600 observations) that were consistent with region three pitch operation.





### 3.1.2 Turbine yaw controller assessment

To ensure optimal rotor alignment, the wind turbine yaw drive orients the rotor plane roughly perpendicular to the turbine inflow wind direction (Mittelmeier and Kühn, 2018). Yaw error is defined as the misalignment angle between the rotor plane and the turbine inflow wind direction (i.e. quantifying the non-normal rotor orientation angle). Between 15:22:00 UTC and 15:31:59 UTC, the $T_T$ controller was instructed to implement a +10° yaw error. Unlike the construct of the blade pitch controller, a wind turbine will not actively yaw on a second-by-second basis to ensure optimal rotor alignment. A wind turbine will typically only yaw when the yaw error has exceeded some threshold (e.g. $\pm10°$) for an extended period of time (e.g. 10 min). As a result, $T_T$ yaw error in the experimental period was expected to deviate from the prescribed +10° offset.

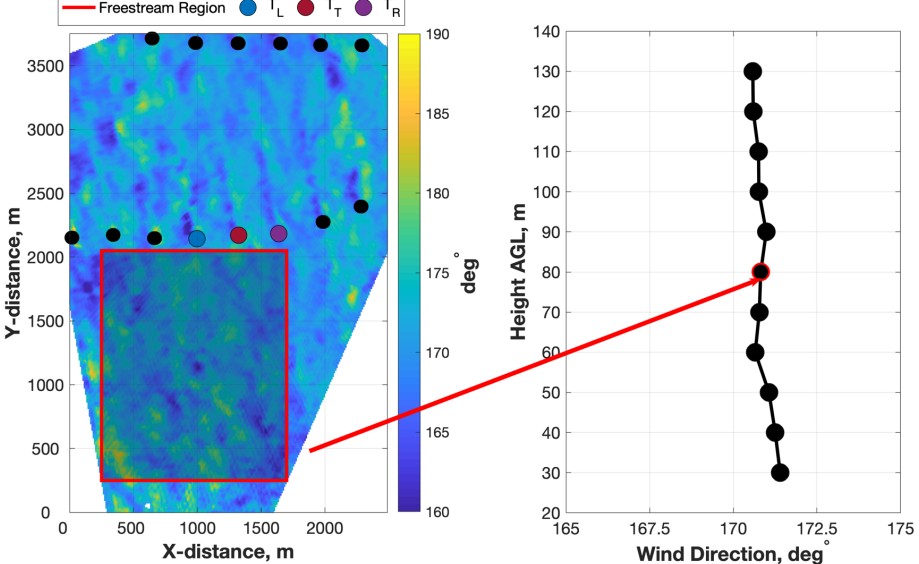

**Figure 5.** (a) TTUKa DD hub-height wind direction (deg°) at 15:31:00 UTC overlaid by the freestream analysis area (red rectangular region). TTUKa DD wind direction measurements within the freestream analysis area were used to determine (b) the mean wind direction at each constant-height plane within the vertical depth of the wind turbine rotor sweep (i.e. the freestream wind direction profile).

Although turbine yaw information was provided, turbine inflow wind direction as defined by the nacelle wind vane was not. Therefore, a freestream wind direction derived from DD radar measurements was used to estimate the turbine inflow wind direction. The freestream wind direction was defined as the mean wind direction measured across a 1.45 km by 1.8 km upstream analysis area (denoted by the red rectangle in Fig. 5a) that was free of obstacles that could impact the flow. The freestream wind direction was defined at each DD constant-height plane within the vertical depth of the wind turbine rotor sweep (Fig. 5b), and the mean of these freestream wind direction measurements was used to determine the rotor sweep area (RSA) average turbine inflow wind direction (i.e. $\theta_{inf}^V$). Due to wind plant and turbine measurement limitations, yaw error is traditionally defined relative to the hub-height wind direction measured by the nacelle wind vane. However, the nacelle wind vane is unable to account for differences in wind direction with height (i.e. wind veer). Therefore, yaw error defined by the



hub-height wind direction will be unable to comprehensively quantify the rotor-sweep relative variations in the axial induction factor that cause wake deflection. Hence, yaw error $\left(\text{i.e. } \theta_{err}^V\right)$ was defined in each DD volume relative to the RSA average turbine inflow wind direction using

$$\theta_{err}^V = \theta_{inf}^V - \theta_{yaw}^V, \qquad\qquad 1$$

where $\theta_{yaw}^V$ was the DD volume yaw angle (i.e. the mean yaw angle during the DD volume acquisition period). Positive magnitudes of $\theta_{err}^V$ denote counterclockwise rotor rotation relative to $\theta_{inf}^V$.

During the +10° experimental period, the $T_T$ exhibited a mean $\theta_{err}^V$ value of +9.43°, a maximum $\theta_{err}^V$ value +14.80°, and a minimum $\theta_{err}^V$ value of +4.60° (Fig. 6). The $T_L$ exhibited a similar mean $\theta_{err}^V$ value of +8.05° during the +10° experimental period, while the $T_R$ exhibited a much smaller mean $\theta_{err}^V$ value of +3.21°. Although the mean $T_T$ $\theta_{err}^V$ value was close to the desired control offset of +10°, there was insufficient evidence to indicate this value was strategic as opposed to just natural variability allowed within the construct of the wind turbine controller. Considering data from the entire DD analysis period (i.e. 14:22:32 UTC – 15:31:57 UTC), the mean $T_T$ $\theta_{err}^V$ value was actually reduced (as opposed to enhanced) in the +10° experimental period. In the DD analysis period, the $T_T$ exhibited a mean $\theta_{err}^V$ value of +13.32°, a maximum $\theta_{err}^V$ value +20.91°, and a minimum $\theta_{err}^V$ value of +4.60°. The $T_L$ and $T_R$ yaw error statistics (i.e. minimum, mean, and maximum $\theta_{err}^V$ values) for the entire DD analysis period are detailed in Table 4.

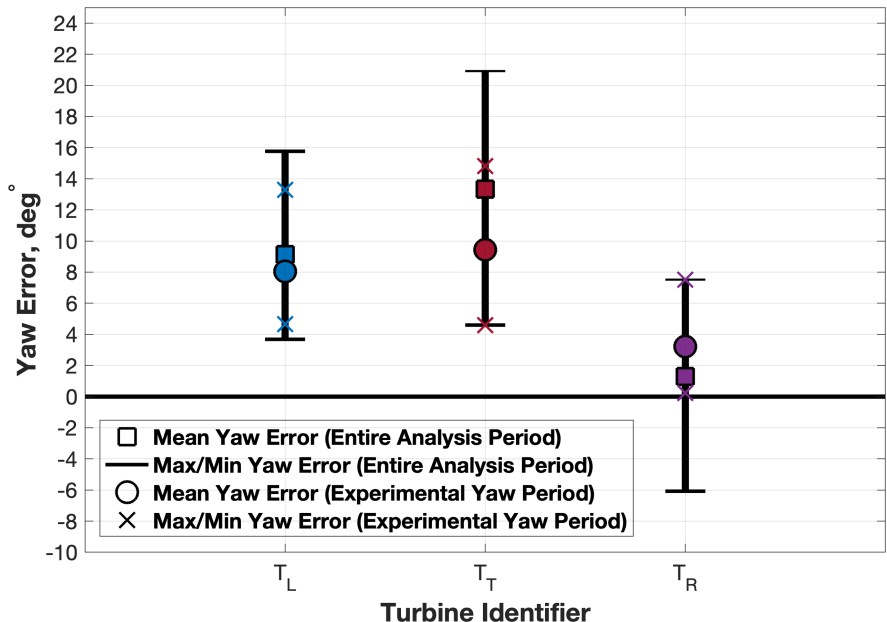

**Figure 6.** The mean, maximum, and minimum $\theta_{err}^V$ values during the +10° experimental period and the entire DD analysis period.





| Turbine Identifier | +10° Experimental Period | | | Data from the Entire DD Analysis Period | | |
|---|---|---|---|---|---|---|
| | Minimum | Mean | Maximum | Minimum | Mean | Maximum |
| $T_L$ | +4.64° | +8.05° | 13.29° | +3.68° | +9.12° | +15.76° |
| $T_T$ | +4.60° | +9.43° | 14.80° | +4.60° | +13.32° | +20.91° |
| $T_R$ | +0.22° | +3.21° | +7.52° | -6.08° | +1.29° | +7.52° |

**Table 4.** The $T_L$, $T_T$, and $T_R$ minimum, mean, and maximum $\theta_{err}^V$ values in the +10° experimental period and within the entire DD analysis period.

### 3.2 Controller assessment challenges

Quantifying wind turbine control changes is imperative to determining the effectiveness of control-based wake mitigation. Being able to accurately quantify the implementation of these wake-mitigating control strategies not only lends insight into their feasibility but also increases confidence that differences in wake structure can be attributed to variations in wind turbine control. An assessment of the $T_T$ controller was performed in Sect 3.1. Analysis determined that in each brief experimental period, the experimental control offsets (i.e. those realized within the individual experimental periods) were less than those prescribed to the $T_T$ controller. Furthermore, based on comparison to the $T_L$ and $T_R$, there was insufficient evidence to indicate the performance of the $T_T$ was significantly modified by implementing the experimental control offsets. However, as detailed below, wind turbine controller assessment was limited by the data and information available and the methods used.

The available inflow information was not optimal for controller assessment. The nacelle anemometer (subject to the NTF) was used to denote the turbine inflow wind speed and define the blade pitch angle offsets. However, nacelle-based measurements are inherently distorted due to their location behind the rotating rotor (Allik et al., 2014). Furthermore, the NTF is a proverbial black box; it is unknown what turbine signals besides the nacelle anemometer are used to produce the turbine inflow wind speed, and also unknown how well it approximates rotor-sweep relative variations in wind speed. Alternatively, area-averaged DD radar measurements were used to denote the turbine inflow wind direction and determine the yaw error. While this wind direction estimate quantifies wind veer, it is a freestream wind direction estimate, and therefore, it is does not necessarily resolve local inflow variability between the $T_L$, $T_T$, and $T_R$. Furthermore, this turbine inflow wind direction estimate was only available at times consistent with the DD volume acquisition period (i.e. every ~60.4 s); this limited the frequency at which wind turbine yaw error could be quantified. Consequently, neither inflow estimate provided a detailed characterization of the turbine inflow conditions at high temporal frequencies. To improve controller assessment, future field campaigns using scanning-based measurements should use advanced analysis techniques (e.g. Duncan et al., 2019), wherein space-to-time conversions are performed on the spatially distributed velocity fields, to provide a comprehensive characterization of the turbine inflow wind speed and direction on a second-by-second basis. These methods were not employed herein because of data availability issues.



Wind turbine controller assessment was also hindered because controller design was not disclosed. Namely, comprehensively quantifying blade pitch control changes requires knowledge of both the region two and region three blade pitch schedules. Albeit analysis of the available SCADA information yielded an estimate of the region three pitch schedule, similar analysis was insufficient to discern the region two pitch schedule. Therefore, only region three SCADA data were considered when

determining the experimental mean blade pitch angle offsets. As a result, the derived blade pitch angle offsets can only be interpreted as a best estimate of the implemented control changes. Furthermore, all that can be established without direct knowledge of the turbine controller are hypotheses detailing why the prescribed control changes were not fully implemented. For example, the brief duration of the experimental periods may have been insufficient to realize/validate the desired control offsets, and the ability to implement the desired control changes might have been impacted by the ABL conditions present

(e.g. region three inflow wind speeds). Both of these factors might have contributed to the experimental control offsets not being fully realized. However, what can be safely assumed is that the feasibility of implementing blade pitch angle changes will be improved for region two inflow, wherein blade pitch is relatively invariable of the turbine inflow wind speed.

## 4 Measuring wind turbine wake response to first-order turbine control changes

While the previous section details several of the challenges associated with quantifying wind turbine controller performance,

some of the same inhibiting factors also limited wake analyses potential. For example, the experimental mean blade pitch angle offsets were only considered in region three. Although a large percentage of the SCADA data in the experimental periods were consistent with region three pitch operation (refer to Table 3 for the percentages in each experimental period), the percentages between the individual turbines (i.e. the $T_L$, $T_T$, and $T_R$) and the experimental periods (i.e. +1°, +2°, and +3°) varied. These percentage differences make it difficult to exclusively attribute variations in wake intensity to the experimental mean blade

pitch angle offsets. Furthermore, the impact of blade pitch on wake structure also depends on the characteristics of the inflow, such as the turbine inflow wind speed. Differences in the turbine inflow wind speed existed between the $T_L$, $T_T$, and $T_R$ and the individual experimental periods (Table 5). Although individual periods could be isolated for comparison of the wake deficits relative to the blade pitch angle offset (e.g. comparing the wake of the $T_T$ and $T_R$ in the +2° experimental period), comprehensively quantifying the impact of blade pitch on wind turbine wake structure requires further data. Future field

campaigns should implement experimental blade pitch control for longer durations to allow the effectiveness of pitch-based wake mitigation to be more comprehensively examined. Extended analysis periods will also enable averaging times sufficiently long to account for natural variability in the wake deficit profile due to turbine inflow variabilities.

| Turbine Identifier | +1° Experimental Period | +2° Experimental Period | +3° Experimental Period | Data from all Experimental Periods |
|---|---|---|---|---|
| $T_L$ | 12.22 | 13.92 | 12.62 | 12.92 |
| $T_T$ | 14.26 | 14.32 | 13.67 | 14.08 |
| $T_R$ | 13.20 | 14.34 | 12.89 | 13.48 |

**Table 5.** The $T_L$, $T_T$, and $T_R$ mean inflow wind speed (m s$^{-1}$) in the +1°, +2°, and +3° experimental periods, as well as using data from each experimental period.



Analysis was therefore limited to examining the impact of yaw error on wake deflection. Because of the brief duration of the +10° experimental period, and also because a wider range of yaw error magnitudes was available by considering data from outside the experimental periods, wake deflection analysis was performed on the entire DD analysis period (i.e. 14:22:32 UTC – 15:31:57 UTC). During the DD analysis period, the $T_L$ exhibited a mean $\theta_{err}^V$ value of +9.12°, the $T_T$ exhibited a mean $\theta_{err}^V$

value of +13.32°, and the $T_R$ exhibited a mean $\theta_{err}^V$ value of +1.29°. Therefore, to examine the impact of larger yaw errors on wake deflection, the wake of the $T_R$ was contrasted against the wakes of both the $T_L$ and $T_T$.

*4.1 DD wake-tracking algorithm*

In order to examine the impact of larger yaw errors on wake deflection, a wake-tracking algorithm (WTA) established by Hirth

and Schroeder (2013) was used to objectively define the wake center location at 0.25-RD intervals from 1 RD to 13 RD downstream. The WTA is detailed below and application of the WTA at 14:59:29 UTC to the wake of the $T_L$, $T_T$, and $T_R$ is demonstrated in Fig. 7.

To resolve the wake of a single wind turbine from the DD measurement volume:

1)    The value of $\theta_{inf}^V$ (defined in Sect. 3.1.2) was used to develop a vertical wake cross-section 1 RD downstream of the turbine (denoted by the horizontal black line in Fig. 7b). The wake cross-section (characterized by a width and height of 1 RD) was oriented normal to the value of $\theta_{inf}^V$ and was vertically centered at hub height. The $T_T$ wake cross-section at 14:59:29 UTC is provided in Fig. 7a.

2)    At each DD constant-height plane within the wake cross-section (i.e. at 10-m intervals between 30 m and 130 m), the wind speed plane was analyzed, and the horizontal location of the wind speed minimum was determined (denoted by the red circles in Fig. 7a).

3)    The median horizontal location of the wind speed minima (denoted by the red square in Fig. 7a) was defined as the

wake center. Vertical wake center meandering was not considered in the WTA, and therefore, the wake was vertically centered at hub height.

4)    Each subsequent wake cross-section was developed directly downwind of the previously derived wake center location. Steps 2 and 3 were repeated at each downstream distance to determine the wake center location outwards of

13 RD downstream. If the wake center location could not be determined at some distance downstream due to data availability, wake tracking for the turbine was discontinued.




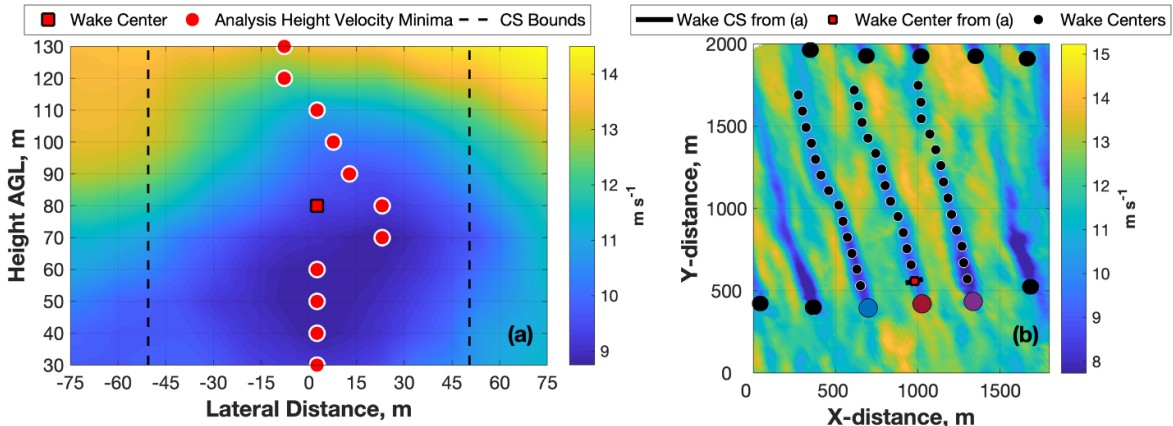

**Figure 7.** (a) $T_T$ wake cross section (CS) wind speed (m s$^{-1}$) at 14:59:29 UTC at 1 RD downstream overlaid by the location of the wind speed minimum at each relevant analysis height (red circles) and the median horizontal location of these minima (i.e. the wake center location [denoted by the red square]). (b) TTUKa DD hub-height wind speed (m s$^{-1}$) at 14:59:29 UTC overlaid by the wake cross-section (black line) and wake center location (red square) from (a). Also shown at 1-RD intervals are the $T_L$, $T_T$, and $T_R$ wake center locations (black circles).

*4.2 Impact of wind turbine yaw error on downstream wake location*

Assuming a uniform inflow wind direction (horizontally and vertically across the wind turbine rotor sweep), the wind turbine wake should extend directly downstream when the rotor plane is oriented perpendicular to the turbine inflow wind direction.

5   However, when the wind turbine exhibits yaw error, wind turbine thrust will vary across the rotor plane causing the wake to be deflected, or skewed, in the direction of the rotor edge facing away from the wind (Fig. 8). Clockwise rotor rotation (i.e. $\theta_{err} < 0$) relative to a fixed inflow wind direction will theoretically induce wake deflection to the left, whereas counterclockwise rotor rotation (i.e. $\theta_{err} > 0$) will theoretically induce wake deflection to the right (Burton et al., 2001).

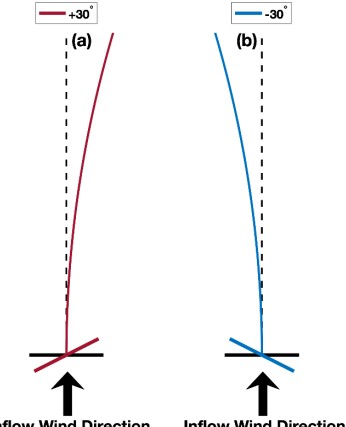

**Figure 8.** Wake deflection theoretically induced by (a) counterclockwise (i.e. $\theta_{err} = +30°$) and (b) clockwise (i.e. $\theta_{err} = -30°$) yaw rotation relative to a fixed inflow wind direction.



To quantify wake deflection due to variations in $\theta_{err}$, a wake skew angle was determined for each wake analyzed. The wake skew angle was defined as

$$\theta_{skew}^V = \theta_{wake}^V - \theta_{inf}^V \qquad\qquad 2$$

where $\theta_{wake}^V$ was the wake centerline angle. Constrained linear least-squares regression (Gill et al., 1981) was used to determine

5   the value of $\theta_{wake}^V$, wherein the wake centerline was required to emanate from the location of the wind turbine. The value of $\theta_{wake}^V$ minimized the error sum of squares; the error distribution was defined as the lateral distance between individual wake center locations and the wake centerline (e.g. Fig. 9a). Positive values of $\theta_{skew}^V$ indicate wake deflection to the right and negative values of $\theta_{skew}^V$ indicate wake deflection to the left. Application of these methods at 15:30:00 UTC to the wake of the $T_L$, $T_T$, and $T_R$ is provided in Fig. 9b.

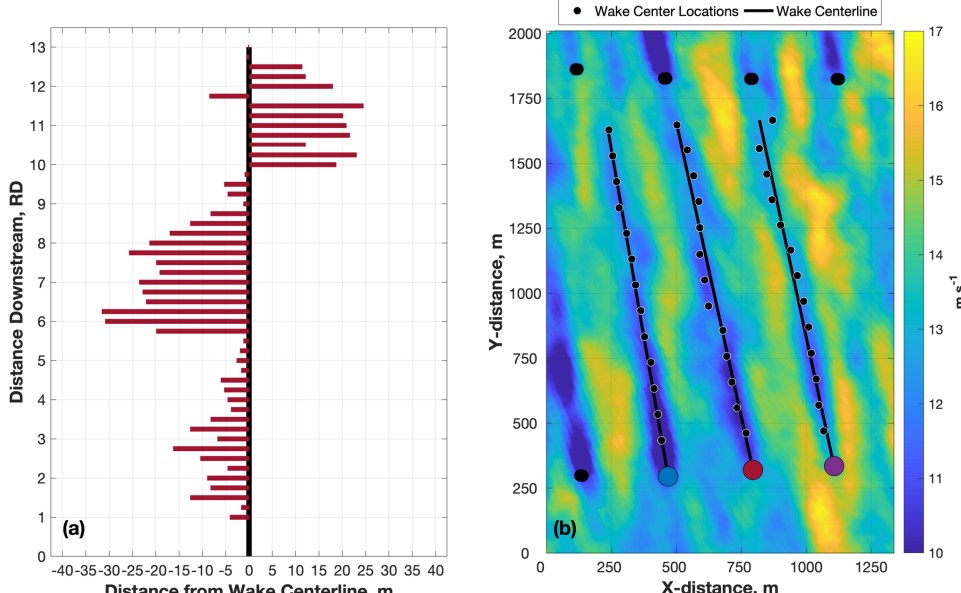

**Figure 9.** (a) The lateral distance between the $T_T$ wake centerline and the wake center locations (i.e. the $T_T$ error distribution). (b) TTUKa DD hub-height wind speed (m s$^{-1}$) at 15:30:00 UTC overlaid by the $T_L$, $T_T$, and $T_R$ wake centerline and the wake center locations at 1-RD increments.

Within the DD analysis period, the wakes of the $T_L$, $T_T$, and $T_R$ were frequently skewed compared to their presumed location based on the value of $\theta_{inf}^V$ (Fig. 10a). However, the value of $\theta_{skew}^V$ did not demonstrate a strong sensitivity to the value of $\theta_{err}^V$. Despite the $T_L$ and $T_T$ exhibiting larger values of $\theta_{err}^V$ than the $T_R$ in the DD analysis period (the $T_L$ exhibited a mean $\theta_{err}^V$ value of +9.12°, the $T_T$ exhibited a mean $\theta_{err}^V$ value of +13.32°, and the $T_R$ exhibited a mean $\theta_{err}^V$ value of +1.29°), all three turbines

15   demonstrated similar magnitudes of $\theta_{skew}^V$ (Fig. 10b). The $T_L$ exhibited a mean $\theta_{skew}^V$ value of -2.90°, the $T_T$ exhibited a mean $\theta_{skew}^V$ value of -3.34°, and the $T_R$ exhibited a mean $\theta_{skew}^V$ value of -2.17° (Fig. 10c). Further demonstrating the limited sensitivity



of $\theta^V_{skew}$ to variations in $\theta^V_{err}$, a linear model fit between $\theta^V_{skew}$ and $\theta^V_{err}$ exhibited a poor $R^2$ value of 0.18. The effectiveness of yaw-based wake steering is expected to be greater at lower wind speeds than at higher wind speeds. This is because wind turbine thrust is reduced at higher wind speeds, as are the rotor-sweep relative variations in the axial induction factor that induce wake deflection. While this can explain why $\theta^V_{skew}$ did not demonstrate a strong sensitivity to $\theta^V_{err}$, it does not explain

5   the average sign (i.e. $\pm$) of $\theta^V_{skew}$ in relation to $\theta^V_{err}$. Despite on average exhibiting positive values of $\theta^V_{err}$, the mean $\theta^V_{skew}$ value was negative for all three turbines; indicating the observed wake deflection (i.e. to the left when looking downstream) was opposite of that expected (i.e. to the right when looking downstream).

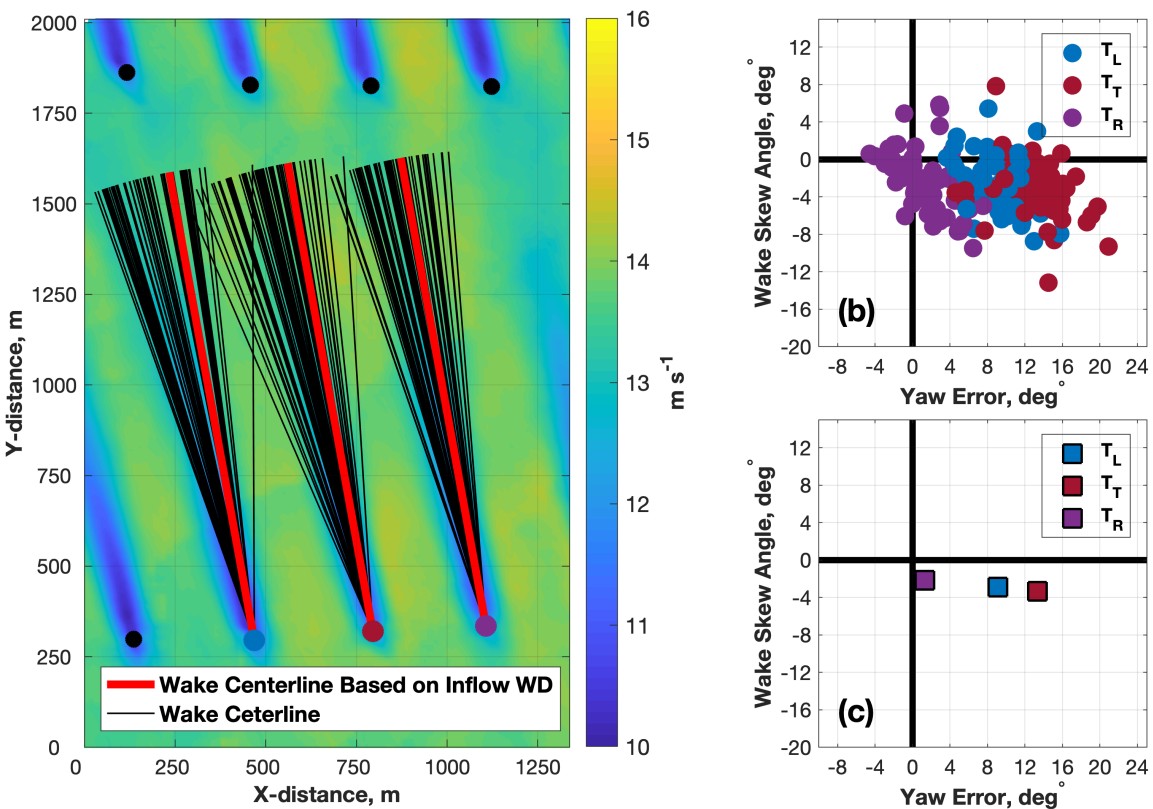

**Figure 10.** (a). Composite mean TTUKa DD hub-height wind speed (m s$^{-1}$) for the DD analysis period overlaid by the $T_L$, $T_T$, and $T_R$ wake centerline from each contributing DD volume (black lines) and the assumed wake centerline based on the DD analysis period mean $\theta^V_{inf}$ value (red lines). (b) The $T_L$, $T_T$, and $T_R$ $\theta^V_{skew}$ values plotted as a function of $\theta^V_{err}$ and (c) the turbine-respective mean values from (b).

Meteorological conditions also impact wind turbine wake structure. Although little is known how local ABL heterogeneities

10   and other coherent turbulent structures modulate the wind turbine wake, previous research suggests downstream wake progression might be modified by transient ABL streak features (Marathe et al., 2016). Therefore, further analysis was performed to examine how ABL streaks, specifically streak orientation, might have promoted the unexpected values of $\theta^V_{skew}$ relative to $\theta^V_{err}$.

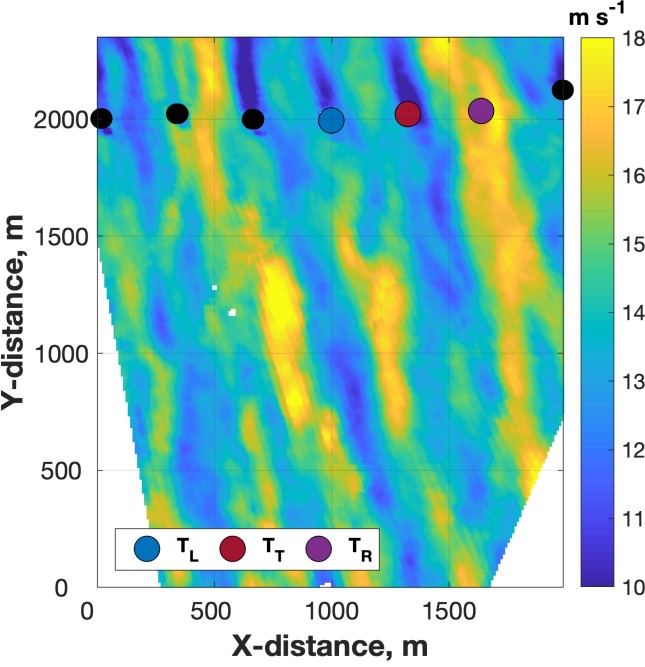

**Figure 11.** TTUKa DD hub-height wind speed (m s⁻¹) at 15:21:08 UTC demonstrating the presence of transient ABL streaks upstream of the lead-turbine row.

### 4.2.1 ABL streak orientation

Boundary layer streaks are elongated, near-surface regions of high- and low-order momentum (e.g. Fig. 11) (Drobinski and Foster, 2003; Träumner et al., 2015). Although these features are elongated in the along-wind dimension, their orientation can

slightly deviate (i.e. clockwise or counterclockwise) from the ABL wind direction (Lorsolo et al., 2008). To examine ABL streak orientation and the extent to which it might have influenced $\theta^V_{skew}$, a method of minimum variance (Lorsolo et al., 2008) was used to quantify streak orientation. Streak orientation was investigated at hub height in each DD volume and analysis was performed in the freestream analysis area to ensure streak orientation was not modified by the wind plant. Furthermore, to facilitate streak characterization, analysis was performed on the residual wind field to ensure the streaks were well resolved.

Residual wind speeds were determined at hub height by removing from the DD wind field the freestream analysis area mean hub-height wind speed (e.g. Fig. 12a).

Streak orientation was determined at hub height in each DD volume by identifying the orientation angle that minimized variance in the residual wind field. In each DD volume, residual wind speed variance was examined across 100 transects, each

13-RD long (i.e. consistent with the search length of the WTA). These transects were orientated at 401 different angles ranging from -20° of $\theta^V_{inf}$ to +20° of $\theta^V_{inf}$ at 0.5° intervals (Fig. 12b). The variance value assigned to each orientation angle was defined as the mean variance measured across the transects (Fig. 12c). Residual wind speed variance was assumed to be minimized in




the direction parallel to the ABL streaks; therefore, the minimum variance value was used to denote the streak orientation angle (i.e. $\theta_{\text{streak}}^{V}$). At 15:21:08 UTC, a $\theta_{\text{streak}}^{V}$ value of 162.72° was determined, which was 9° counterclockwise (i.e. to the left when looking downstream) of $\theta_{\text{inf}}^{V}$. However, within the DD analysis period, the value of $\theta_{\text{streak}}^{V}$ was on average offset of $\theta_{\text{inf}}^{V}$ by +1.85°. The difference between $\theta_{\text{streak}}^{V}$ and $\theta_{\text{inf}}^{V}$ (i.e. $\theta_{\text{streak}}^{V} - \theta_{\text{inf}}^{V}$) is hereinafter denoted by $\theta_{\text{S-skew}}^{V}$.

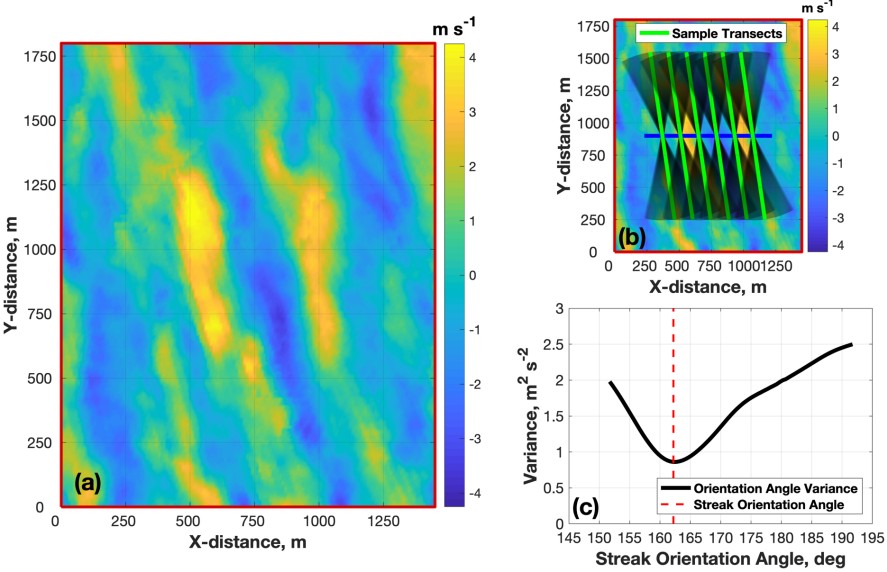

**Figure 12.** (a) TTUKa DD hub-height residual wind field (m s⁻¹) within the freestream analysis area at 15:21:08 UTC. (b) Same as (a) except overlaid by sample transects and the range of streak orientation angles examined (denoted by the black shaded regions). (c) The mean variance value assigned to each streak orientation angle; the minimum of this variance profile denotes the $\theta_{\text{streak}}^{V}$ value of 162.72° at 15:21:08 UTC.

The mean sign of $\theta_{\text{S-skew}}^{V}$ (i.e. +) was opposite of the turbine-respective mean $\theta_{\text{skew}}^{V}$ signs (i.e. -) in the DD analysis period. However, despite this difference, the peak of both the $\theta_{\text{S-skew}}^{V}$ and $\theta_{\text{skew}}^{V}$ distributions occurred between -5° and 0° (Fig. 13). ABL streaks were more prevalent in some DD volumes than in others, but regardless of their presence, the method of minimum variance always extracts a streak orientation angle. It is possible the orientation angles assigned in periods of reduced streak prevalence might have impacted the mean value (and specifically the sign) of $\theta_{\text{S-skew}}^{V}$. Nevertheless, similarity in the peak of the $\theta_{\text{S-skew}}^{V}$ and $\theta_{\text{skew}}^{V}$ distributions is more noteworthy than the slight difference in the mean value of the distributions, and suggests ABL streak orientation might have contributed to the unexpected values $\theta_{\text{skew}}^{V}$ relative to $\theta_{\text{err}}^{V}$. Furthermore, the advection direction of local ABL heterogeneities, such as ABL streaks, has previously been shown to be counterclockwise (i.e. to the left when looking downstream) of the local mean wind direction (Duncan et al., 2019).



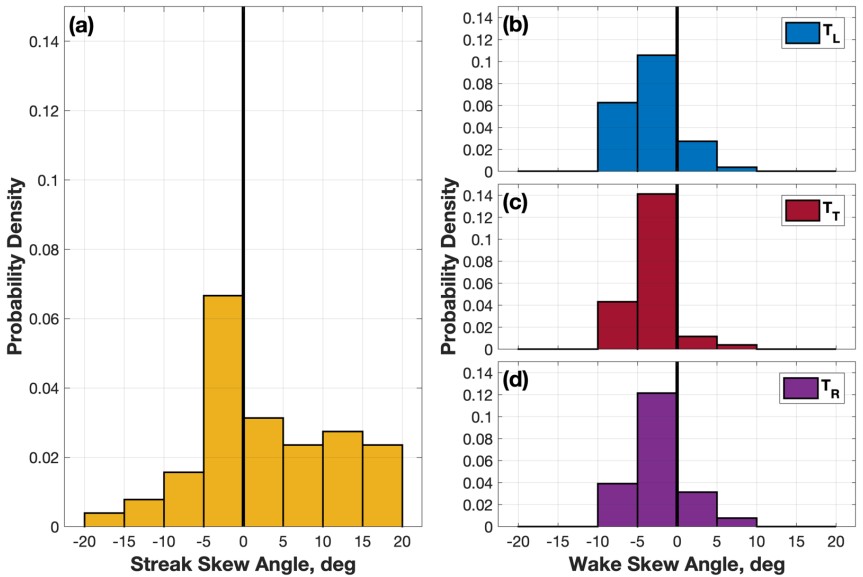

**Figure 13.** The probability density of the (a) $\theta^V_{\text{S-skew}}$ distribution and the (b) $T_L$, (c) $T_T$, and (d) $T_R$ $\theta^V_{\text{skew}}$ distributions.

Although the prevalence of ABL streaks might have contributed to the ineffectiveness of yaw-based wake steering, there was insufficient evidence to conclude they are the root cause of the mean sign of $\theta^V_{\text{skew}}$. Regardless, these results are important because they suggest simply implementing yaw error might not be enough to ensure effective wake steering; rather, an

integrated knowledge of ABL heterogeneities and their characteristics is needed (e.g. interaction with these transients might amplify or inhibit wake deflection). These results demonstrate the importance of research that examines the impact of transient ABL heterogeneities and coherent turbulent structures on wind turbine wake structure and variability. This information is needed to optimize wind plant control.

## 5 ABL stability driven wake changes

To better understand wind turbine wake response to changes in the ABL, and how this might impact the effectiveness of control-based wake mitigation, wind turbine wake evolution (i.e. wake length and meandering) within the EET was examined. The EET denotes the transition between the daytime convective and nocturnal stable ABLs. This evolutionary period of atmospheric stability is consistent with distinct changes in wind speed, wind direction, wind structure, atmospheric turbulence, and temperature (Mahrt, 1981; Nieuwstadt and Brost, 1986; Acevedo and Fitzjarrald, 2001; Edwards et al., 2006). To track

the progression of the EET and the onset of the nocturnal stable ABL, the virtual potential temperature (i.e. $\theta_v$) gradient between 10 m and 200 m was examined using available meteorological tower data. The gradient in $\theta_v$ gives an indication of how conducive the ABL is towards the development of turbulent eddies, and therefore, can be used to discern atmospheric stability. The ABL was defined as stable when the $\theta_v$ gradient was positive and was defined as unstable when the $\theta_v$ gradient was negative. The $\theta_v$ gradient was determined for each SD sector scan by analyzing thermal measurement from a 10-min



period that was temporally centered on the sector scan completion time (i.e. the scan-centric analysis period). Using these scan-centric $\theta_v$ gradient values, the convective ABL lasted from the start of data collection (22:32:08 UTC) to 23:27:53 UTC, and the stable ABL persisted until the conclusion of data collection (00:52:30 UTC) (Fig. 14).

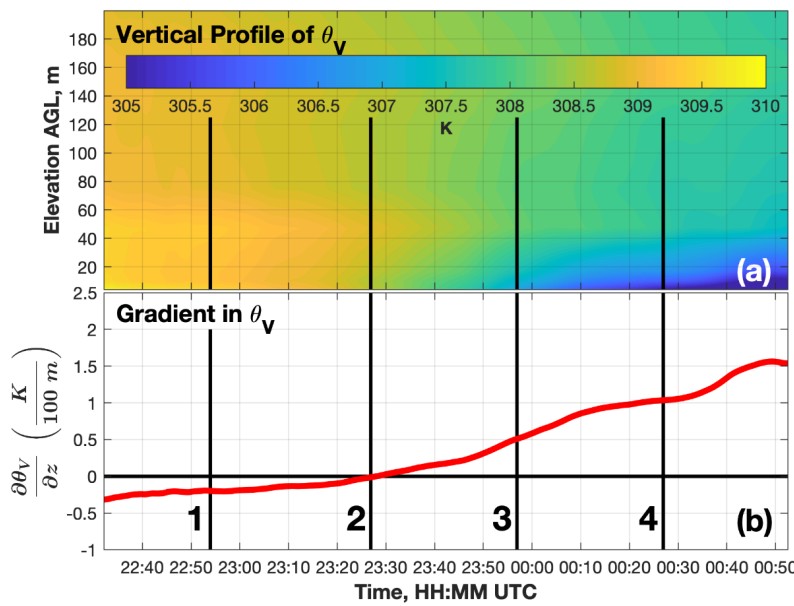

**Figure 14.** Evolution during the SD data acquisition period of the scan-centric (a) $\theta_v$ vertical profile (K) and (b) the $\theta_v$ gradient (K per 100 m). The magnitude of the $\theta_v$ gradient at the times consistent with the numerically labelled vertical black lines is provide in Table 6.

Wind characteristics evolved with the progression of the EET. Mean wind speeds derived from the scan-centric analysis periods at the 74.7 m meteorological tower level were strongest (i.e. exhibiting a mean value of 8.16 m s$^{-1}$) in the convective ABL before steadily decreasing in magnitude to a mean value of 7.48 m s$^{-1}$ in the nocturnal stable ABL (Fig. 15a). To characterize changes in the intensity of atmospheric turbulence, trends in both turbulence intensity (TI) and turbulence kinetic energy (TKE) were examined (Fig. 15b). TI was defined as the ratio between the mean ($\mu$) and standard deviation ($\sigma$) $\left( \text{i.e.} \frac{\sigma}{\mu} \right)$

of wind speed measurements made within the individual scan-centric analysis periods, and TKE was defined as $0.5(\sigma_u^2 + \sigma_v^2 + \sigma_w^2)$ where $\sigma$ with subscripts u, v, and w denote the standard deviation of the along-wind, lateral, and vertical wind field components, respectively. Both TI and TKE exhibited their peak values in the convective ABL before decreasing in magnitude with the progression of the nocturnal stable ABL. Values of the $\theta_v$ gradient, mean wind speed, TI, and TKE at various times within the SD analysis period (denoted by the numerically labelled black lines in Figs. 14 and 15) are provided in Table 6. Measurements

at 74.7 m AGL were analyzed because of their proximity to the wind turbine hub height.





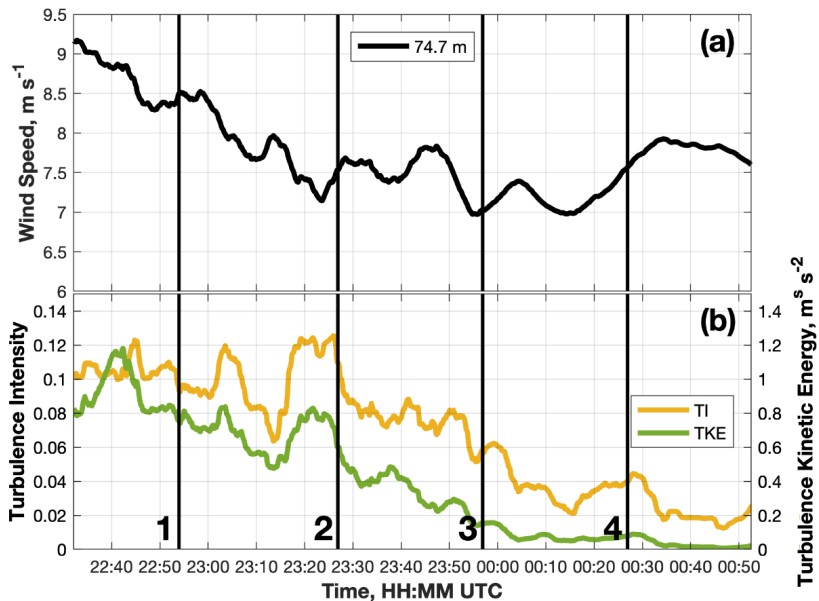

**Figure 15.** Evolution during the SD data acquisition period of the scan-centric (a) mean wind speed (m s$^{-1}$) and (b) both TI and TKE at 74.7 m AGL. The scan-centric mean wind speed, TI, and TKE values at the times consistent with the numerically labelled vertical black lines are provided in Table 6.

| ABL Stability and Turbulence Parameters | (1) 22:53:57 UTC | (2) 23:26:53 UTC | (3) 23:56:54 UTC | (4) 00:26:56 UTC |
|---|---|---|---|---|
| $\theta_v$ Gradient (K per 100 m) | -0.20 | -0.01 | 0.51 | 1.04 |
| Wind Speed (m s$^{-1}$) | 8.50 | 7.54 | 7.03 | 7.57 |
| TI | 0.09 | 0.11 | 0.06 | 0.04 |
| TKE (m$^2$ s$^{-2}$) | 0.76 | 0.60 | 0.15 | 0.08 |

**Table 6.** 12 October 2015 SD deployment ABL characterization summary. Provided at (1) 22:53:57 UTC, (2) 23:26:53 UTC, (3) 23:56:54 UTC, and (4) 00:26:56 UTC (i.e. corresponding to the numerically labelled vertical black lines in Figs. 14 and 15) is the $\theta_v$ gradient and at 74.7 m AGL the mean wind speed, TI, and TKE.

*5.1 SD WTA*

Slight modifications were made to the DD WTA to accommodate SD measurement of the wind turbine wake. These modifications are detailed below.

1)    Because SD sector scans were performed at a one-degree elevation tilt, analysis of the vertical wake structure at incremental distances downstream was not possible. The SD wake cross-sections were instead developed along the sloped 2D plane (i.e. roughly defined for a single height AGL at each distance downstream) and the wake center location was defined as the horizontal location of the wake cross-section wind speed minimum. However, to facilitate derivation of wake length (defined in Sect. 5.2), the width of the wake cross-section was increased to 200 m.



2) SD measurements do not resolve wind direction. Therefore, both the downstream bearing and normal orientation of the wake cross-sections were defined by a 60-s scan-centric (i.e. ±30 s of the sector scan completion time) average wind direction derived using tower measurements at 74.7 m AGL.

3) Because the wind turbine RD cannot be disclosed, the WTA was not applied at RD-normalized increments (i.e. 1 RD, 1.25 RD, etc.), but instead was applied at 50-m intervals between 150 m and 3200 m downstream.

Application of the SD WTA to the wind turbine wake at 22:38:25 UTC is provided in Fig. 16a.

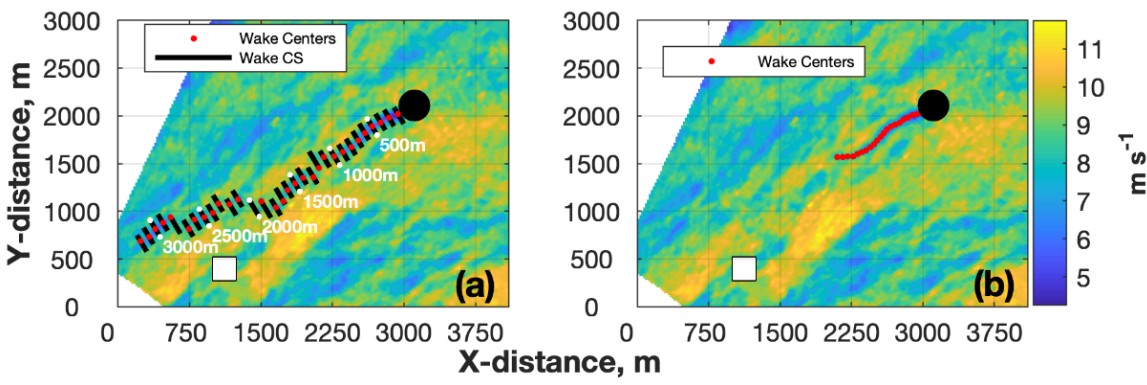

**Figure 16.** (a) The WTA applied to the TTUKa radial velocity wind field (m s⁻¹) at 22:38:25 UTC overlaid by the location of the instrumented meteorological tower (white square) and the wind turbine (black circle). Also plotted at 100-m intervals are the individual wake cross-sections (black lines) and wake center locations (red circles). (b) Valid wake center locations after applying the wake cessation criteria.

*5.2 Wake length*

The size and intensity of turbulent eddies impact the rate at which the wake-adjacent wind field (i.e. the freestream) is mixed into the wake. Variations in the magnitude of turbulence-induced mixing between the convective and stable ABLs should cause wake length to grow with the onset and progression of the EET. However, the WTA does not differentiate between wake

and non-wake (e.g. transient lull features) velocity minima. Provided adequate data availability, wake center locations were always defined to a downstream distance of 3200 m. To discern wake cessation, the lateral shift between successive wake center locations, defined as the distance between the actual and projected wake center location, was analyzed. Wake cessation was defined as the downstream distance where the lateral shift between successive wake center locations exceeded 50 m. The objective when developing these methods was to discern trends in wake length. Therefore, these values should be interpreted

as robust estimates rather than exact measures of wake length. Application of these methods at 22:38:25 UTC produced a wake length of 1000 m (Fig. 16b).





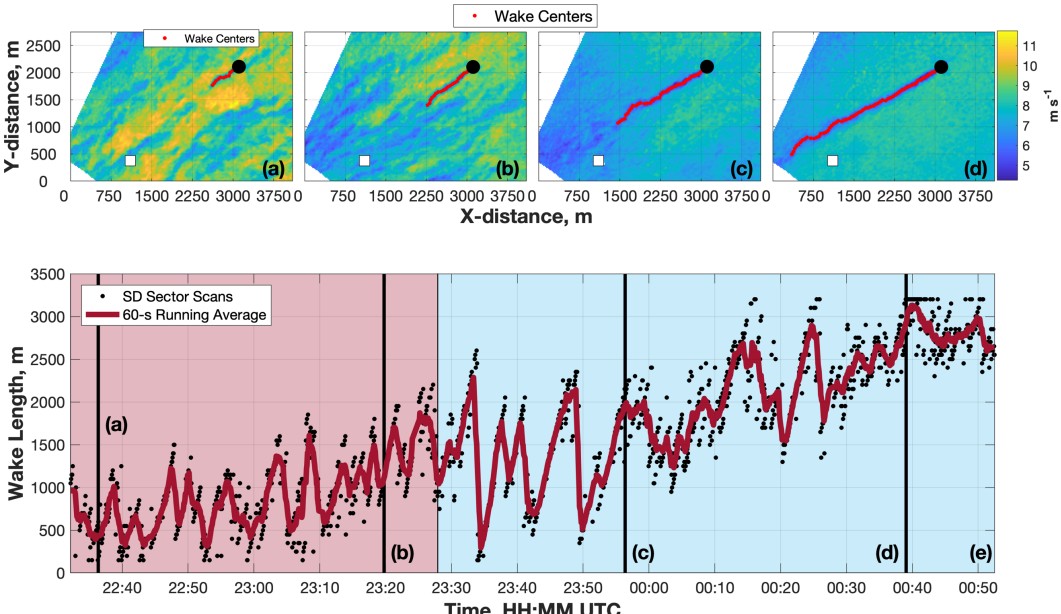

**Figure 17.** The TTUKa radial velocity wind field (m s⁻¹) at (a) 22:36:21 UTC, (b) 23:19:46 UTC, (c) 23:56:23 UTC, and (d) 00:39:04 UTC overlaid by the location(s) of the instrumented meteorological tower (white square), the wind turbine (black circle), and the WTA-derived wake centers (red circles). (e) Wake length time history overlaid by a 60-s running average of wake length; the red and blue shaded regions denote the convective and stable portions of the SD data acquisition period, respectively.

Clear variations in wake length existed between the convective (e.g. at 22:36:21 UTC [Fig. 17a] and 23:19:46 UTC [Fig. 17b]) and stable (e.g. at 23:56:23 UTC [Fig. 17c] and 00:39:04 UTC [Fig. 17d]) ABLs. In the convective ABL, wake length fluctuated between 150 m (i.e. the lowest detectable wake length) and 2200 m about a mean value of 907 m. The prevalence of turbulent transients in the convective ABL likely impacted the ability of the turbine to continuously impart optimal thrust, and therefore, contributed to the absence of a persistent downstream wake. As the EET progressed and ABL stability increased, wake length began to grow. However, this trend was not steady, rather large and temporally sharp changes in wake length occurred (Fig. 17e) at times. For example, at 22:33:50 UTC a portion of the wake became 'detached' from the turbine (Fig. 18a). The detached wake feature advected downstream in subsequent sector scans allowing a new contiguous wake to form (Fig. 18b). Similar wake discontinuities occurred at 23:38:13 UTC, 23:41:00 UTC, and 23:49:01 UTC. These discontinuities can be attributed to suboptimal wind turbine thrust and demonstrate the importance of wake analyses at smaller time scales. Much of this variability would be lost if the measurement periods were increased and/or the analysis was averaged across longer time frames. Despite these discontinuities, wake length increased by an average of 115.92 % between the convective and stable ABLs.



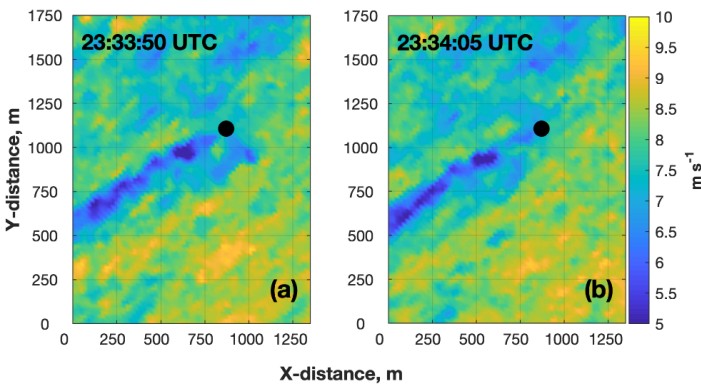

**Figure 18.** The TTUKa radial velocity wind field (m s$^{-1}$) at (a) 23:33:50 UTC and (b) 23:34:05 UTC demonstrating wake discontinuities.

### 5.3 Wake meandering

Wake meandering is the oscillating behaviour of the wake in both the horizontal and vertical dimensions of the atmosphere as it advects downstream. Wake center variability about the wake centreline was analyzed to discern variations in wake
5  meandering consistent with the onset and progression of the EET. However, only horizontal wake meandering was considered because the SD strategies exclusively provided measurements along a single sloped 2D plane. Linear regression was used to determine the wake centerline, albeit for this analysis the wake centerline was not required to emanate from the location of the wind turbine (Fig. 19a). The mean ($\mu$) and standard deviation ($\sigma$) of the wake center variability (i.e. lateral variability about the wake centreline [e.g. Fig. 19b]) was used to quantify horizontal wake center meandering at incremental distances
10  downstream. Horizontal wake center meandering statistics (i.e. $\mu \pm 1\sigma$) are provided in Fig. 20 as a function of ABL stability. Statistics were only plotted for downstream distances where at least 50 wake centers were derived (wake centers downstream of the wake cessation point were not analyzed).

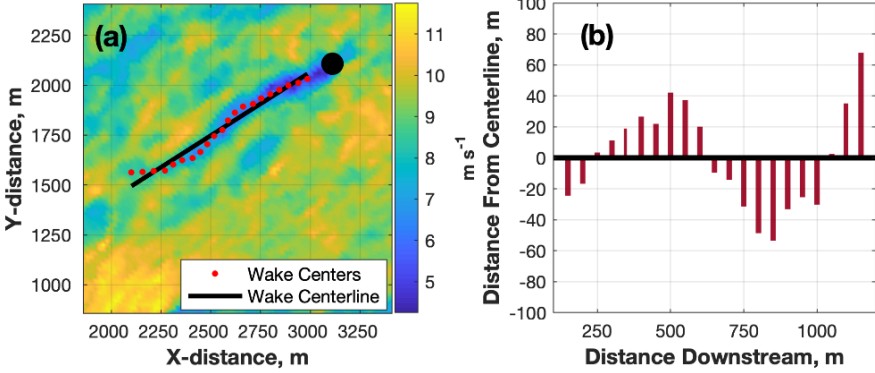

**Figure 19.** (a) The TTUKa radial velocity wind field (m s$^{-1}$) at 22:38:25 UTC overlaid by the wind turbine (black circle), the wake centerline (black line), and the individual wake center locations (red circles). (b) Vertical red lines denote the lateral deviation (m) of the wake centers from the wake centerline.



The σ value of horizontal wake center meandering was amplified in the convective ABL. For downstream distances where wake meandering statistics were defined for both the convective and stable ABLs (i.e. from 150 m to 1600 m), $\sigma_{stable}$ was on average 37.55% smaller than $\sigma_{convective}$, and at no common distance downstream was $\sigma_{stable}$ greater than $\sigma_{convective}$. In both the convective and stable ABLs, the value of σ initially decreased with distance downstream (until approximately 500 m) before steadily increasing in magnitude. Despite this initial decrease, which might be due to deepening velocity deficits, the value of $\sigma_{convective}$ increased by 82.02% from a mean value of 25.64 m at 150 m downstream to a mean value of 46.67 m at 1000 m downstream. Alternatively, only a 27.80% increase was noted between these two distances in the stable ABL.

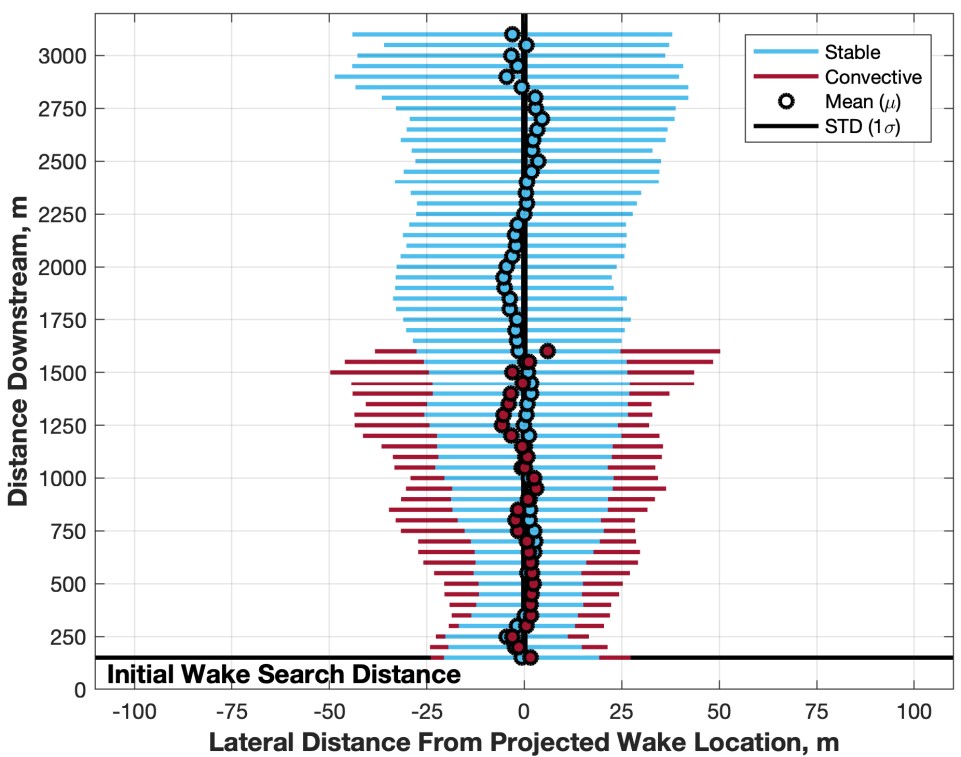

**Figure 20.** Horizontal wake center meandering mean values (colored circle) and one standard deviation bounds (colored horizontal line) as a function of distance downstream (the plotted colors reflect the ABL stability classification).

## 6 Concluding remarks

Wake measurements were made using the TTUKa radars to examine the effectiveness of two wake-mitigating control strategies at full-scale. However, instead of validating these wind plant control methods, results highlighted some of the complexities associated with executing and analysing wind plant control at full-scale using brief experimental control periods. Some difficulties include (1) the ability to accurately implement the desired control changes, (2) identifying reliable data sources and methods to allow these control changes to be accurately quantified, and (3) attributing variations in wake structure




to turbine control changes rather than a response to the underlying atmospheric conditions (e.g. boundary layer streak orientation, atmospheric stability).

Modifications were made to the yaw and blade pitch of a utility-scale wind turbine (i.e. the $T_T$) in this study to examine how
control changes impact wind turbine wake structure and downstream progression. However, despite instructing the $T_T$ controller to implement distinct blade pitch angle and turbine yaw offsets, in each experimental period the observed control offsets were less than those prescribed. Furthermore, based on comparison to the $T_L$ and $T_R$, there was insufficient evidence to indicate the performance of the $T_T$ was significantly modified by implementing the experimental control offsets. These results indicate a reactive wind turbine controller might struggle to accurately implement turbine control offsets, at least for the time
scales (i.e. 10 min) and wind speeds (i.e. region three) examined. However, wind turbine controller assessment was limited; the available inflow information was not optimal for controller assessment and without direct access to controller design, it was impossible to determine why the control offsets were not fully realized. Based off these results, comprehensive controller assessment requires (1) a data source providing near-continuous, and reliable (i.e. accurate) turbine inflow information, and (2) access to controller design so any factors inhibiting proper implementation of the turbine control offsets can be identified.
However, if this information cannot be secured, the presented methods can be used as a reference guide for controller assessment.

Because of experimental difficulties, analysis of wind turbine wake response to turbine control changes was limited to examining the impact of yaw error on wake deflection. The wakes of the $T_L$, $T_T$, and $T_R$ were frequently skewed compared to
their presumed location based on the value of $\theta_{inf}^V$. The mean direction of $\theta_{skew}^V$ for all three turbines (i.e. to the left when looking downstream) was opposite of that expected (i.e. to the right looking downstream) based on the values of $\theta_{err}^V$. To examine whether local boundary layer transients such as ABL streaks might be modifying downstream wake progression, ABL streak orientation was quantified and compared to $\theta_{skew}^V$. Although the mean value of $\theta_{S\text{-skew}}^V$ (i.e. $\theta_{streak}^V - \theta_{inf}^V$) was different than the turbine-respective $\theta_{skew}^V$ values, the peak of both the $\theta_{S\text{-skew}}^V$ and turbine $\theta_{skew}^V$ distributions was between -5°
and 0°, which suggests ABL streak orientation might have contributed to the unexpected values $\theta_{skew}^V$ relative to $\theta_{err}^V$. This result is important because it indicates proactively implementing yaw error might not be sufficient to ensure effective wake steering without at least an integrated knowledge of ABL heterogeneities and their characteristics.

To further examine how atmospheric conditions might impact the effectiveness of wind plant control, wind turbine wake
evolution within the EET was examined. Between the convective and stable ABLs, mean wake length increased by 115.92%, indicating the likelihood of negative turbine-to-turbine wake interaction will be amplified in the stable ABL. Furthermore, the σ value of horizontal wake center meandering was reduced by 36.55 % in the stable ABL. This reduction in σ indicates the downstream deterministic location of the wake should be increased in the stable ABL, which should also increase the

effectiveness of control-based wake modulation (e.g. wake steering). These results are significant and suggest the potential benefit and feasibility of wind plant control should be increased in the stable ABL, where both the potential for wake-related power losses and the ability to modify the wake should be amplified. In other ABL environments, local ABL heterogeneities might govern wake structure, variability, and downstream progression. More research is needed to understand how, in addition

to turbine controls, transient ABL heterogeneities impact and govern wind turbine wakes. This information is fundamental to optimizing wind plant control.

*Data availability*

The data are not publicly available because of the agreed upon non-disclosure agreements with the turbine manufacturer.

*Author contributions*

James Duncan, Brian Hirth, and John Schroeder contributed to the design and execution of the two radar deployments. Data analyses were performed by James Duncan under the supervision of both Brian Hirth and John Schroeder. The manuscript was prepared by James Duncan and reviewed by both Brian Hirth and John Schroeder.

*Competing interests*

The authors do not have any competing conflicts of interest.

*Acknowledgements*

Funding for this study was provided by the National Science Foundation award CBET-1336935. The authors also thank Dr. Richard Krupar III for assistance with the data collection efforts and Jerry Guynes for preparing the TTUKa radars for deployment.



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
