# Peer review of "Exploring the complexities associated with full-scale wind plant wake mitigation control experiments"

_Wind Energy Science, 2019_

## Referee Comment (RC1) · Anonymous Referee #1 · 1 Dec 2019

This paper discusses two measurement campaigns: 1) a wind plant control experiment involving changes in pitch offset and yaw misalignment at a commercial wind plant with wake measurements from Doppler radars, and 2) a measurement campaign where wake length and the degree of wake meandering were characterized using Doppler radar measurements for different atmospheric stability conditions. The paper discusses the inability to enact the desired control changes in the first experiment, explaining the shortcomings, and discusses how inhomogeneities in the wind flow can make it difficult to distinguish control impacts on wake behavior from the impacts of atmospheric variations. The second measurement campaign highlights how the length of wakes increases in stable conditions while the amount of meandering decreases, and the

authors explain how this means wind plant control should be more effective in stable conditions.

The paper is written very clearly and is easy to follow. And the paper shows how valuable Doppler radar measurements can be in wind plant control experiments. However, one area that I believe needs to be improved is the explanation of the goal of the wind plant control experiment and how the analysis presented connects to the goal. The paper focuses on analysis of data from an experiment, but it isn't clear what the original objectives were. Was the goal to measure changes in wake behavior, or to look at the impact on power of the downstream turbines? Why were only a few 10-minute control periods used rather than a longer experiment that would more clearly reveal trends? Given the original objectives of the experiment, what would the authors do differently next time?

Furthermore, the authors should connect this work to the existing literature on field validation of wind plant control concepts. More of a review of previous work in the field should be provided and the authors should discuss how the objective of their work fits in with what has already been published, rather than saying that previous work "remains limited." Is there a gap the authors are trying to address with this research?

In addition, there are several areas where explanations and methods should be improved, as explained in the comments below.

Specific comments:

Pg. 2, ln. 17: Double check your references listed. For example, Vollmer et al. 2016 and Fleming et al., 2018 are listed as wind tunnel experiments, but these are numerical simulations.

Pg. 2, ln. 22: Another recent full-scale validation of wind plant control is: Howland et al., Wind farm power optimization through wake steering, Proceedings of the National Academy of Sciences, 2019.

Pg. 2, ln. 25: "To expand upon existing full-scale validation efforts, agreements were made with an industry partner..." Please be sure to review the existing full-scale validation efforts and explain how the present work fits in.

Section 2.2: Can you explain if the 12 October experiment is at a different site? Are both sites in similar terrain, or are there significant differences between them that should be pointed out?

Fig. 1: Please explain the meaning of the different colors of wind turbines, including white, to avoid confusion.

Pg. 6, ln. 1: What purpose do the downstream turbines (white circles) serve in this experiment?

Pg. 6, ln. 18: Fleming et al., 2018 deals with numerical simulations, do you mean 2019?

Pg. 7, ln. 6: If the benefit of modifying blade pitch is greater in region 2, then why was the sole half-hour experiment period in region 3? Would it have made more sense to wait for more favorable conditions?

Pg. 7, ln. 9: "To maintain the rated generator speed in region three, the wind turbine follows a pitch schedule to extract the desired amount of momentum at various wind speeds." Blade pitch controllers typically use generator speed feedback to control blade pitch to regulate generator speed. Therefore, if you are adding a pitch offset in region three, what else are you changing in the controller so that the pitch controller doesn't simply compensate for the offset to bring the generator speed back to rated? Is the generator torque or gen. speed setpoint also changed? Could it be that the pitch offset that is added is simply an offset to the "fine pitch" (minimum pitch) angle that the turbine operates at below rated, and that there is no real change to the pitch control above rated? More detail about the intended pitch offset strategy would be helpful.

Pg. 7, ln. 12: "The region three pitch schedule was constructed by fitting a linear model

to the distribution of blade pitch angles..." Pitch schedules are generally very nonlinear as a function of wind speed, especially near rated wind speed. Can you elaborate on your choice of a linear pitch schedule model?

Pg. 9, ln. 12: A 1.45 km x 1.8 km averaging area seems too large for determining the local inflow wind direction to the turbines, especially if you are trying to distinguish between the wind inflow to each of the three turbines. Furthermore, given the advection time across the 1.8 km analysis area, the estimated wind directions are likely not very well correlated with what the turbines see at a high temporal resolution. Can you try this with 100 m x 100 m averaging areas, local to each turbine? This could improve your results, or at least make them more meaningful.

Section 3.1.2: Given the large positive mean yaw errors with or without the offset applied, it seems possible that the yaw position reported in the SCADA data is not calibrated properly. It is common for yaw position values from SCADA data to deviate significantly from the true orientation (i.e., 0 degrees -> true north) over time. Was the calibration of the yaw position data confirmed? If not, this should be discussed further.

Pg. 11, ln. 15: "However, nacelle-based measurements are inherently distorted..." Another factor to consider is that the flow distortion from the rotor can change as the control changes. Adding a pitch offset could cause the wind speed behind the rotor to change differently than with the original control, complicating the detection of changes in turbine operation as a function of wind speed.

Pg. 12: ln. 10: "Both of these factors might have contributed to the experimental control offsets not being fully realized." Certainly for yaw control, a single 10-minute period might not be sufficient to observe meaningful yaw misalignment changes, given the slow dynamics of yaw controllers.

Section 4.1: I would suggest a revised wake tracking algorithm in light of improvements in the understanding of wake deflection physics. As discussed in papers such as the following, yaw misalignment can cause wakes to have a "curled" shape due to the presence of counter rotating vortices. This means the peak velocity deficit could change with height and averaging across all heights in the rotor disk area is not necessarily the most relevant metric.

-Vollmer et al., Estimating the wake deflection downstream of a wind turbine in different atmospheric stabilities: an LES study, Wind Energy Science, 2016.

-Howland et al., Wake structure in actuator disk models of wind turbines in yaw under uniform inflow conditions, Journal of Renewable and Sustainable Energy, 2016.

-Fleming et al., A simulation study demonstrating the importance of large-scale trailing vortices in wake steering, Wind Energy Science, 2018.

A more meaningful lateral wake center estimate for wind plant control applications can be found using the method explained in Vollmer et al. 2016, where the cubed wind speed is averaged across a hypothetical rotor disk area centered at different lateral displacements. The displacement that results in the lowest value can be considered the wake center position.

Fig. 9: Is Fig. 9 (a) showing the distance from the centerline of the wake after correcting for the skew angle, or from the centerline in the mean wind direction? Please clarify what is being shown.

Pg. 16, ln. 6: "...indicating the observed wake deflection... was opposite of that expected" How might wind veer impact the wake deflection during the experiment period? Could this be an explanation for the unexpected skew?

Pg. 18, ln. 3: "9 degrees counterclockwise... of theta^V_inf." Stating what theta^V_inf is would clear up any confusion about the sign convention.

Section 4.2.1: The potential impact of streak orientation on wake skew angle is an interesting idea. However, a deeper discussion of how this might cause the skewing of the wake would be appreciated.

Fig. 13: Consider showing the joint probability density of streak skew angle and wake skew angle. This would support your idea of a correlation between the two better.

Section 5.1: SD WTA: For a section title, consider spelling out the acronym.

Section 5: In addition to wake length and wake meandering, what differences have you observed in the relative magnitude of the velocity deficits for stable vs. unstable conditions? This would be a valuable addition to the paper.

Pg. 26: ln. 14: "access to the controller design so any factors inhibiting proper implementation of the turbine control offsets can be identified." I agree that access the controller improves the assessment of wind plant control strategies, and is always desirable, but I think meaningful control assessments can be done without direct access. For example, adding a pitch offset in region 2 (where pitch is typically fixed at "fine pitch") could be achieved without needing to understand the controller dynamics. Furthermore, to implement a yaw misalignment, the yaw controller setpoint could be changed from zero to the desired offset, but full understanding of the controller dynamics is not necessary, and in many cases would be asking too much given the proprietary nature of wind turbine control systems.

---

## Referee Comment (RC2) · Anonymous Referee #2 · 9 Dec 2019

Thank you for this paper. It is very useful to receive experimental results, and it is always a major undertaking to gather such data. In general the paper is well written with good and useful figures.

My major criticism/suggestion is that the results presented in sections 3/4 are of too little data, and with too many "black box" issues to be used to draw conclusions from. Section 5 on the other hand provides results which are useful, line up with physical interpretation, and show show statistical significance. I would therefore propose to condense (or remove?) sections 3/4, and perhaps expand a bit on section 5.

Specific comments:

[Figure]

Wake centerline analysis: I might propose this analysis is not necessarily the most interesting. For one thing, this would really only apply to the averaged wake, and I don't believe you have enough data obtained (the dynamic wake can look very different from the average). Further, considering that much recent research implies that wake steering changes the structure of the wake, in addition to deflecting it, (c.f. Howland, Michael F., et al. "Wake structure in actuator disk models of wind turbines in yaw under uniform inflow conditions." Journal of Renewable and Sustainable Energy 8.4 (2016): 043301.) or (Martínez-Tossas, et al. The aerodynamics of the curled wake: a simplified model in view of flow control. United States: N. p., 2019. Web. doi:10.5194/wes-4-127-2019.) These suggest that defining the centerline can become very challenging, also given the presence of cross-flows generated in steering. A more straightforward calculation could be the speed of the flow behind the turbine in the direction of the expected wake direction, looking for variation there.

P 2 "operate below their peak capacity to decrease wake effect..." this describes well static induction control, but less well wake steering and dynamic induction control

Fig 1: It's explained later, but the legend is unclear in meaning, perhaps explain more in caption

P6: RD?

Bottom p9: Could alternatively define wind direction as the average yaw position of non-changed turbines?

Fig 8: Believe these wake directions are convex in the wrong direction, the wake deflection appears to be accelerating as heading downstream, whereas expectation would be recovery to main direction (cf fig 1 in Jiménez, Ángel, Antonio Crespo, and Emilio Migoya. "Application of a LES Technique to Characterize the Wake Deflection of a Wind Turbine in Yaw." Wind Energy, 2010. https://doi.org/10.1002/we.) this might also impact analysis in fig 10

P 19 "simply implementing yaw error might not be enough to ensure effective wake steering" not clear this result can be drawn from these results

Fig 17 and Fig 20: Great figures and really interesting!! Text analysis also interesting

---

## Author Comment (AC1) · 28 Jan 2020

James B Duncan Jr
https://doi.org/10.5194/wes-2019-78-RC1

**General Response**

Thank you for your comments and the time taken to provide this review. Please find in this document a response to your specific comments. The authors are confident the responses provided will clear up any confusion or concerns regarding the methods used. Please find as supplemental material an edited version of the manuscript with changes tracked.

Foremost, the authors would like to state that at the time of the experiment (14 December 2014), experimental validation of wind plant control at full scale was rare. Similarly, there was a dearth of observational data (especially compared to numerical simulation) characterizing in three-dimensions wind turbine wake structure and variability. Therefore, in collaboration with an industry partner, the preliminary objectives of the experiment were to (1) examine three-dimensional wind turbine wake response to changes in wind turbine yaw and blade pitch and (2) examine how these changes impact the net power production of individual turbines in the wind plant (i.e. quantifying the effectiveness of the wind plant control strategy). However, experimental logistics (e.g. experimental control limitations imposed by the wind plant operator) ultimately inhibited execution of these experimental objectives. Therefore, experimental objectives and analysis focus evolved to exploring the complexities and difficulties associated with performing a wind plant control experiment at full scale. The results of this study lend insight into these difficulties, and therefore, should be used to inform future field campaigns. The authors are confident the changes made in response to your comments allow the manuscript to tell a more direct story.

**Specific Comments**

**Pg. 2, ln. 17: Double check your references listed. For example, Vollmer et al. 2016 and Fleming et al., 2018 are listed as wind tunnel experiments, but these are numerical simulations.**

Vollmer et al. (2016), Fleming et al. (2018), and also Park and Law (2016) were based on numerical simulation. By mistake, these works were initially miscited in the manuscript as studies based on wind tunnel experiments. Furthermore, although Jiménez et al., (2010) leverages the results of previously performed wind tunnel experiments, a significant portion of the manuscript was based on numerical simulation. The citing of these manuscripts was appropriately modified in the manuscript and the authors are confident that the referenced works are now properly cited.

The text now reads (Pg. 2 Lns. 15 through 20):
> "The benefit of wind plant control has been previously demonstrated using numerical simulation (e.g. Jiménez et al., 2010; Johnson and Fritsch, 2012; Lee et al., 2013; Annoni et al., 2015; Park and Law, 2015; Fleming et al., 2015; Gebraad and van Wingerden, 2015; Gebraad et al., 2016; Park and Law, 2016; Vollmer et al., 2016; Fleming et al., 2018; Kanev et al., 2018) and in wind tunnel experiments (e.g. Parkin et al., 2001; Corten and Schaak, 2003; Howland et al., 2016; Schottler et al., 2017; Bartl et al., 2018; Bastankhah and Porté-Agel, 2019)."

Furthermore, the reference for Fleming et al. (2019) has been updated to reflect its status being upgraded from 'In Review' to 'Published'.

**Pg. 2, ln. 22: Another recent full-scale validation of wind plant control is: Howland et al, Wind farm power optimization through wake steering, Proceedings of the National Academy of Sciences, 2019.**

Thank you for bringing this work to our attention, it is now cited in the manuscript (Pg. 2 Ln 19). Furthermore, based on your input the authors discovered another relevant study by Howland wherein wind tunnel experiments were used to examine the impact of yaw operation on wake structure. This work (documented below) is now referenced in the manuscript.

Howland, M. F., Bossuyt, J., Martínez-Tossas, L. A., Meyers, J., and Meneveau, C.: Wake structure in actuator disk models of wind turbines in yaw under uniform inflow conditions, Renew. Energ. Sustain. Dev., 8, 043301, 2016.

**Pg. 2, ln. 25: "To expand upon existing full-scale validation efforts, agreements were made with an industry partner..." Please be sure to review the existing full-scale validation efforts and explain how the present work fits in.**

As mentioned in the manuscript, full-scale validation of wind plant control remains limited. Despite the potential benefit of wind plant control being demonstrated in both numerical simulation and wind tunnel experiments, full-scale validation of these control techniques must be performed before wind plant control can be commercially employed.

Based on your comment, the authors recognize that the phrasing,

> "To expand upon existing full-scale validation efforts, agreements were made with an industry partner to modify the yaw and blade pitch of a utility-scale wind turbine for a limited time period to examine the resulting variations in wake structure."

is vague when it comes to defining exiting full-scale validation efforts, and also does not denote how the presented research expects to contribute to the research field.

Existing experimental validation of wind plant control at full-scale has nominally focused on either (1) quantifying the benefit of wind plant control by analyzing the power and controls data of individual turbine pairs in a wind plant over extended periods (e.g. Fleming et al., 2017a; Ahmad et al., 2019; van der Hoek et al., 2019; Fleming et al., 2019; Howland et al., 2019) or (2) using advanced measurement technologies (e.g. lidar) to examine near-wake response to turbine control changes (Trujillo et al., 2016; Fleming et al., 2017b).

The referee should consider the date of the experiment (14 December 2014) when factoring in how this research fits into the now currently published research. The initial focus of the manuscript was not to necessarily build on any specific experimental field campaigns (very few experimental studies were published at this time), but rather to contribute to the general dearth of full-scale experimental datasets examining wind plant control and its efficacy. The preliminary objectives

of this experiment were to (1) examine three-dimensional wind turbine wake response to changes in wind turbine yaw and blade pitch in both the near- and far-wake regions and (2) examine how these changes impact the net power production of individual turbines in the wind plant (i.e. quantifying the effectiveness of the wind plant control strategy). However, experimental objectives evolved based on the experimental difficulties encountered. Analysis focus ultimately evolved to exploring the complexities and difficulties associated with performing a wind plant control experiment at full scale.

However, the authors recognize that the principal objectives of this experiment were not properly reflected in the manuscript by simply stating 'To expand upon existing full-scale validation efforts...'. Therefore, the manuscript text was modified to (1) briefly highlight the state of experimental wind plant control validation at full-scale and (2) to better describe the experimental objectives of the 14 December 2014 experiment. The authors are confident the modified text (copied below) more accurately defines the scope and contents of the manuscript.

Pg. 2 Lns. 24 through 34 and Pg. 3 Lns 1 through 2:
> "Experimental validation of wind plant control at full-scale has frequently relied upon the analysis of power and controls data from individual turbine pairs in a wind plant to quantify the benefit of various wind plant control techniques (e.g. Fleming et al., 2017a; Ahmad et al., 2019; van der Hoek et al., 2019; Fleming et al., 2019; Howland et al., 2019). However, few studies have used advanced measurement technologies (such as lidar or radar) to document differences in wake structure due to the turbine control changes implemented as part of wind plant control (e.g. Trujillo et al., 2016; Fleming et al., 2017b). Additionally, these studies almost exclusively limit wake measurement to the near-wake region, and therefore, are unable to monitor the downstream progression of these control-induced wake modifications. To contribute to these full-scale validation efforts, and to expand the downstream extent to which control-induced wake changes are measured, agreements were made with an industry partner to modify the yaw and blade pitch of a utility-scale wind turbine for a limited time period to examine the resulting variations in wake structure. Wake measurements were made using Texas Tech University's Ka-band (TTUKa) Doppler radars employing dual-Doppler (DD) scanning strategies. However, rather than validating the effectiveness of these wind plant wake-mitigating control strategies, results highlight some of the complexities associated with executing and analysing wind plant control strategies at full-scale using brief experimental periods."

**Section 2.2: Can you explain if the 12 October experiment is at a different site? Are both sites in similar terrain, or are there significant differences between them that should be pointed out.**

The 14 December 2014 and 12 October 2015 radar experiments were performed at different sites. This is in part why instrumented tower data was available for the 12 October 2015 SD radar deployment but not for the 14 December 2014 DD radar deployment. However, both sites were located in the US Great Plains and there were no significant terrain differences between the two sites that would impact wind plant complex flow structure or variability. Although confidentiality agreements preclude disclosure of the radar deployment locations, manuscript text was modified

to provide more details regarding the general location (i.e. the US Great Plains) of the two radar deployments.

Pg. 3 Lns. 23 through 24 now reads:
> "The technical specifications of the TTUKa radars are further detailed in Table 1 and radar deployment specifics (both performed in the US Great Plains) are provided in the subsections below."

**Fig. 1: Please explain the meaning of the different colors of wind turbines, including white, to avoid confusion.**

A brief description of the turbines was provided in the in the initial manuscript submission in the first paragraph of Section 3.

Pg. 5 Lns. 11 through 16 and Pg. 6 Lns. 1 through 2:
> "Located in the DD domain of the 14 December 2014 deployment were 20 wind turbines distributed across two turbine rows. The wind turbines were characterized by a hub height of 80 m and a rotor diameter (RD) of 101 m. Supervisory control and data acquisition (SCADA) information detailing the turbine inflow wind speed (subject to the nacelle transfer function [NTF]), turbine yaw orientation, and blade pitch angle were provided at a one-hertz sampling frequency from 14:00:00 UTC to 16:59:45 UTC for seven of the wind turbines (denoted by the non-black circles in Fig. 1). Three of the seven wind turbines were located in the leading row of the wind plant, while the remaining four were located in the trailing row. The three lead-row wind turbines were separated by an average distance of 1512.2 m (~15 RD) from the trailing turbine row and were laterally separated from each other by an average distance of 321.1 m (~3 RD)."

However, the authors recognize that this text is disconnected from the description of the 14 December 2014 radar deployment (i.e. Section 2.1) and Fig. 1. Therefore, this text was moved to the end of Section 2 and was also slightly modified to better explain the figure (in particular the meaning of the different color turbines). Furthermore, both the caption and legend to Fig. 1 were modified to improve reader comprehension. The revised text is copied below for reference.

Pg. 4 Lns. 14 through 23:
> "Located in the DD domain of the 14 December 2014 deployment were 20 wind turbines distributed across two turbine rows. The wind turbines were characterized by a hub height of 80 m and a rotor diameter (RD) of 101 m. Supervisory control and data acquisition (SCADA) information detailing the turbine inflow wind speed (subject to the nacelle transfer function [NTF]), turbine yaw orientation, and blade pitch angle were provided at a one-hertz sampling frequency from 14:00:00 UTC to 16:59:45 UTC for seven of the wind turbines (denoted by the non-black circles in Fig. 1). Three of the seven wind turbines were located in the lead row of the wind plant (denoted by the blue, red, and purple circles in Fig. 1), while the remaining four were located in the trailing row (denoted by the white circles in Fig. 1). The three lead-row wind turbines were separated by an average distance of 1512.2 m (~15 RD) from the trailing turbine row and were laterally separated from each other by an average distance of 321.1 m (~3 RD). The wake of these

three lead-row wind turbines (referred to as the $T_L$, $T_T$, and $T_R$) were analyzed to examine the effectiveness of the implemented wake-mitigating control strategies."

**Pg. 6, ln. 1: What purpose do the downstream turbines (white circles) serve in this experiment.**

SCADA data from the four downstream turbines were used along with SCADA data from the three lead-row wind turbines to estimate the region three pitch schedule used in analyses. This was indicated in Sect. 3.1.1 of the initial manuscript submission.

Pg. 7 Lns 13 through 16:
> "Data from all seven wind turbines were used to ensure a robust estimate of the region three pitch schedule (a total of 41,343 SCADA wind speed and blade pitch angle measurements were used); measurements inconsistent with region three pitch operation and $T_T$ data from the experimental periods were not considered when constructing the pitch schedule."

However, to improve reader comprehension, the manuscript text was slightly modified to that copied below.

Pg. 7 Lns 29 through 32:
> "Data from all seven wind turbines with SCADA information provided (i.e. the non-black circles in Fig. 1) were used to ensure a robust estimate of the region three pitch schedule (a total of 41,343 SCADA wind speed and blade pitch angle measurements were used); measurements inconsistent with region three pitch operation and $T_T$ data from the experimental periods were not considered when constructing the pitch schedule."

**Pg. 6, ln. 18: Fleming et al., 2018 deals with numerical simulations, do you mean 2019?**

Thank you for noticing this, the text was appropriately modified to Fleming et al. (2019).

**Pg. 7, ln. 6: If the benefit of modifying blade pitch is greater in region 2, then why was the sole half-hour experiment period in region 3? Would have it made more sense to wait for more favorable conditions?**

The wind plant control experiment would have been ideally performed for extended experimental durations and in an environment more conducive to the effectiveness of the implemented control strategies. However, experimental logistics ultimately inhibited execution of the experiment when conditions were optimal. These logistical difficulties are detailed below.

Foremost, although the TTUKa radars can comprehensively document wind plant complex flow structure and variability at high spatial resolutions, data availability is dependent on the atmospheric conditions present (namely the size distribution of scatterers and aerosols present in the ABL). This dependency is also true for other remote sensing instrument such as lidar. Given the specifications of the TTUKa radars, data availability is enhanced in certain precipitating

environments. Therefore, wind plant control experiments were limited to periods when the TTUKa radars were expected to have sufficient data availability. Further impacting rapid radar deployment when atmospheric conditions were ideal was the relative proximity of the wind plant to the staging (i.e. storage) location of the radars. The wind plant was located several hundred miles away from the TTUKa radars when not in use. Therefore, accurate forecasts of atmospheric conditions conducive to data availability had to be made with sufficient lead time to (1) enable proper coordination with the wind plant operators (control experiments were not permitted at all times) and (2) to deploy the radars to the wind plant site. Successful execution of the wind plant control experiment was therefore not as simple as waiting for conditions to become ideal. Furthermore, periods conducive to data availability were not necessarily correlated with atmospheric conditions conducive to the effective implementation of wind plant control (as demonstrated in the manuscript).

However, the 14 December 2014 deployment was not the only attempt at performing a wind plant wake mitigation control experiment. Wind plant control experiments were also performed at a wind plant elsewhere in the US Great Plains and at the TTU Scaled Wind Farm Technology (SWiFT) facility (albeit this is a scaled wind farm facility). The 14 December 2014 radar deployment was presented in this manuscript in part because the wind plant operator allowed a relatively wide range of experimental turbine control changes to be implemented. Prior to recent years, some wind farm operators were reluctant to collaborate on these types of experiments for fear that implementing wind plant wake-mitigating control techniques might impact turbine warranty.

Finally, this field experiment, albeit performed in somewhat of a suboptimal environment and having inherent limitations, lends insight into the complexities associated with performing a full-scale wind plant control experiment. The authors are confident this manuscript adds value to the scientific community and can be used to inform future field campaigns.

**Pg. 7, ln. 9: "To maintain the rated generator speed in region three, the wind turbine follows a pitch schedule to extract the desired amount of momentum at various wind speeds." Blade pitch controllers typically use generator speed feedback to control blade pitch to regulate generator speed. Therefore, if you are adding a pitch offset in region three, what else are you changing in the controller so that the pitch controller doesn't simply compensate for the offset to bring the generator speed back to rated? Is the generator torque or generator speed setpoint also changed? Could it be that the pitch offset that is added is simply an offset to the "fine pitch" (minimum pitch) angle that the turbine operates at below rated, and that there is no real change to the pitch control above rated? More detail about the intended pitch offset strategy would be helpful.**

Due to the proprietary nature of the information, the construct of the wind turbine controller was not provided by the turbine manufacturer. Therefore, the authors are unable to expand on the blade pitch angle offset strategy, nor are they able to state with confidence why these changes were not effectively implemented. This is reflected in the manuscript on Pg. 7 Lns. 3 through 5:

> "Not having access to wind turbine controller design is a major challenge to fully understanding wind turbine behaviour (Fleming et al., 2019), or rather, how the controller

responds to variable inflow conditions when attempting to enact the desired control offsets. Therefore, the provided discussions do not detail why the turbine was able or unable to enact the desired control changes, but rather focuses on quantifying the resulting offsets."

However, it was the author's understanding that the blade pitch angle offsets (i.e. +1°, +2°, +3°) were implemented independent of the inflow state and operation of the turbine (i.e. region two or three).

The referee indicates that generator speed (i.e. rpm) is typically used to help regulate wind turbine blade pitch. Therefore, even if blade pitch were directly modified by incorporating the blade pitch angle offset, the turbine might recognize inconsistent generator speeds relative to the turbine inflow wind speed and appropriately modify the blade pitch. However, regardless of whether generator speed was relied upon to help regulate blade pitch, experimental mean blade pitch angle offsets were not negligible (i.e. Figs. 3 and 4 of the manuscript). Experimental period (i.e. +1°, +2°, +3°) generator speed behavior was also examined to try and gain a more comprehensive understanding of the implemented control changes. However, analyses did not lend much insight, and therefore, these analyses were not incorporated into the manuscript.

**Pg. 7, ln. 12: "The region three pitch schedule was constructed by fitting a linear model to the distribution of blade pitch angles..." Pitch schedules are generally very nonlinear as a function of wind speed, especially near rated wind speed. Can you elaborate on your choice of a linear pitch schedule model?**

Pg. 7 Lns. 29 through 32:
"Data from all seven wind turbines with SCADA information provided (i.e. the non-black circles in Fig. 1) were used to ensure a robust estimate of the region three pitch schedule (a total of 41,343 SCADA wind speed and blade pitch angle measurements were used); measurements inconsistent with region three pitch operation and $T_T$ data from the experimental periods were not considered when constructing the pitch schedule."

The constructed region three blade pitch schedule is provided below (note: due to confidentiality agreements the blade pitch angle values could not be plotted along the y-axis). In this figure, there is a rough linear relationship between blade pitch and the region three turbine inflow wind speeds. However, detracting from this linear relationship is a small magnitude of clustering in blade pitch between 12 and 13 m s$^{-1}$. This clustering is believed to occur as a result of a rapid increase (i.e. at timescales less than the response time of the blade pitch controller) in the turbine inflow wind speed from velocities consistent with region two to region three. Within this period, the turbine is unable to appropriately modify its blade pitch angle to be consistent with region three pitch operation. While it would be desirable to remove these measurements, these values are surrounded by blade pitch angle measurements consistent with region three pitch operation and only represent 1.19 % of the total number of measurements considered. Therefore, this clustering is not expected to significantly impact the construction of the region three pitch schedule.

[Figure]

The authors are confident that these methods enable a robust best-estimate of the region three pitch schedule. Further confidence can be placed on these methods based on the results of Duncan et al. (2019).

Duncan, J. B., Hirth, B. D., and Schroeder, J. L.: Enhanced estimation of boundary layer advective properties to improve space-to-time conversion processes for wind energy applications, Wind Energ., 22, 1203-1218, 2019.

In Duncan et al. (2019), this pitch schedule was used to estimate variations in generator speed due to suboptimal blade pitch angle activity.

**Pg. 9, ln. 12: A 1.45 km by 1.8 km averaging area seems too large for determining the local inflow wind direction to the turbines, especially if you are trying to distinguish between the wind inflow to each of the three turbines. Furthermore, given the advection time across the 1.8 km analysis area, the estimated wind directions are likely not very well correlated with what the turbines see at a high temporal resolution. Can you try this with 100 m x 100m averaging areas, local to each turbine? This could improve your results, or at least make things more meaningful.**

The referee contends a 1.45 km by 1.8 km averaging area is too large to discern turbine-specific inflow differences. Therefore, the referee argues the value of $\theta_{\text{inf}}^{V}$ will likely not be well correlated with the turbine inflow wind direction at higher temporal resolutions. In order to more accurately resolve turbine inflow conditions, the referee recommends using a 100 m by 100 m averaging area local to each turbine. However, due to the interpretation of the DD synthesized wind fields, this 100 m by 100 m turbine-local averaging area is also not necessarily appropriate. Denoted on Pg. 3

Ln. 30 and Pg. 12 Ln 9, the TTUKa radars take on average 60.4 s to collect a single DD volume, and within this DD volume acquisition period, turbulent structures move. Therefore, denoted in Sect. 2.1, Pg. 4 Lns. 5 through 7:

> "The DD wind maps can be interpreted as a pseudo-average of the wind conditions over the volume acquisition period, where the DD volume time stamp denotes the end of the volume period."

Space-to-time conversion techniques (e.g. Duncan et al., 2019) are required to accurately resolve temporal inflow variability at sub-volume timescales (i.e. less than the measurement revisit period). Using local measurements without a proper understanding of how turbulent structures move within the DD volume acquisition period can confound results. Ideally, space-to-time conversions would have been used in Sect. 3.2 to develop a semi-continuous stream of turbine inflow information (wind speed and direction) over the DD volume acquisition period. This would have allowed for temporal inflow variability to more accurately defined, and therefore, would have enabled a more comprehensive determination of the implemented control changes. Denoted on Pg. 12 Lns. 11 through 16:

> "To improve controller assessment, future field campaigns should place precedence on turbine inflow measurements (wind speed and direction) independent of the turbine control system (i.e. non-SCADA data). Experiments using scanning-based measurements should use advanced analysis techniques, such as those established in Duncan et al. (2019) wherein space-to-time conversions were performed on the spatially distributed velocity fields, to provide a comprehensive characterization of the turbine inflow wind speed and direction on a second-by-second basis. Application of these methods was limited because of data availability issues."

These methods were not employed because of data availability issues. While a more local turbine inflow estimate (specifically laterally) might have slightly improved result accuracy, the use of a larger-scale averaging area does not reduce the meaningfulness of the results. Given the average time between DD volume scans (i.e. 60.4 s) and the mean wind speed within the DD analysis period (i.e. 13.71 m s$^{-1}$ as defined by the mean wind field within the freestream analysis area), a relatively large averaging area is not a bad estimate of the mean conditions observed at the turbine over the DD volume acquisition period. While the freestream analysis area is still larger than the longitudinal wind field area that might advect through the turbine over the DD volume acquisition period (based on Taylor's hypothesis this would be a distance of 828.1 m [i.e. 60.4 s x 13.71 m s$^{-1}$]), differences between these two inflow estimates are not expected to be significant.

Regardless, analysis was performed to demonstrate the sensitivity of the turbine inflow estimate to the averaging area used. Comparisons were made between the turbine inflow wind direction determined using (1) the freestream analysis (i.e. $\theta_{inf}^{V}$) and (2) local turbine averaging areas representing the estimated portion of the upstream wind field that will advect through the turbine over the subsequent volume acquisition period based on Taylor's hypothesis. A comparison of these different inflow estimates at 15:31:00 UTC is demonstrated in the figure below. The left subplot demonstrates the spatial dimensions of the individual averaging areas at 15:31:00 UTC, while the right subplot demonstrates the turbine inflow wind direction profile derived from these

averaging areas. The profile-averaged turbine inflow wind direction is provided in the legend of the right subplot for each averaging area. Despite analyzing significantly different areas (especially laterally), differences between the turbine inflow wind direction estimates were small. At 15:31:00 UTC, the $T_L$ inflow wind direction was 0.7° greater than the value of $\theta_{inf}^V$, the $T_T$ inflow wind direction was equal to the value of $\theta_{inf}^V$, and the $T_R$ turbine inflow wind direction was 1.2° less than the value of $\theta_{inf}^V$.

[Figure]

The same analysis was performed for each DD volume in the analysis period to demonstrate that these small differences were not rare. Time histories of $\theta_{inf}^V$ and the local turbine inflow estimates are provided in the figures below for each turbine. The black lines denote the $\theta_{inf}^V$ time history and the colored lines denote the local turbine inflow estimates. The local turbine averaging areas were redefined in each DD volume using Taylor's hypothesis based on the mean conditions in the freestream analysis area. While Taylor's hypothesis was used to give a general idea of the sensitivity of the turbine inflow estimate to the averaging area used, the referee should refer to Duncan et al. (2019) for why Taylor's hypothesis cannot be systematically relied upon to accurately denote the advection of the upstream wind field and hence why these techniques were not used in the manuscript.

Provided in the legend of each figure are both the mean deviation and the mean absolute deviation between $\theta_{inf}^V$ and the local turbine inflow estimates for the entire DD analysis period. Time history data gaps are due to data availability issues; at least 50 % of the respective averaging area was required to determine the mean wind direction estimate. Considering inflow estimate data from all turbines and DD volumes, the mean deviation between the two turbine inflow estimates (the local turbine inflow estimate minus the value $\theta_{inf}^V$) was -0.2° and the mean absolute deviation was 1.1°. These results do not indicate that the use of the freestream analysis area to estimate the turbine

James B Duncan Jr
https://doi.org/10.5194/wes-2019-78-RC1

inflow wind direction (i.e. $\theta_{inf}^{V}$) reduced the meaningfulness of the results. However, the authors do acknowledge that analyses would have benefited from more accurate turbine inflow estimates.

[Figure]

[Figure]

[Figure]

Furthermore, the use of the freestream analysis area to estimate the turbine inflow wind direction provides a common reference frame for defining both the wake (i.e. $\theta_{skew}^{V}$) and streak (i.e. $\theta_{S-skew}^{V}$) skew angles.

**Section 3.1.2: Given the large positive mean yaw errors with or without the offset applied, it seems possible that the yaw position reported in the SCADA data is not calibrated properly. It is common for yaw position values from SCADA data to deviate significantly from the true orientation (i.e., 0 degrees -> true north) over time. If not, this should be discussed further**.

The referee raises the concern that yaw calibration errors might have impacted results because improper yaw calibration is not uncommon. Although not discussed in the manuscript, analysis was performed to discern whether a yaw calibration error was present in either the $T_L$, $T_T$, or $T_R$.

Foremost, regardless of any yaw calibration error, variations in yaw error (i.e. $\theta_{err}^{V}$) should elicit distinct changes in wake skew (i.e. $\theta_{skew}^{V}$). Counterclockwise rotor rotation relative to a fixed turbine inflow wind direction (i.e. a net positive increase in $\theta_{err}^{V}$) should enhance wake deflection to the right when looking downstream (i.e. resulting in a net increase in $\theta_{skew}^{V}$), and clockwise rotor rotation relative to a fixed turbine inflow wind direction (i.e. a net negative decrease in $\theta_{err}^{V}$) should enhance wake deflection to the left when looking downstream (i.e. resulting in a net decrease in $\theta_{skew}^{V}$). However, demonstrated in Fig. 10b and the figure below, it is tough to discern any distinct trends in $\theta_{skew}^{V}$ resulting from variations in $\theta_{err}^{V}$. Based on the figure below, it appears as though the value of $\theta_{skew}^{V}$ slightly decreased as the value of $\theta_{err}^{V}$ increased. However, when data from all three turbines is considered (i.e. Fig. 10b), and thereby a wider range of $\theta_{err}^{V}$ values considered, this relationship was not as apparent. Therefore, yaw calibration errors cannot explain the general insensitivity of in $\theta_{skew}^{V}$ to variations in $\theta_{err}^{V}$.

James B Duncan Jr
https://doi.org/10.5194/wes-2019-78-RC1

[Figure]

Alternatively, assume each turbine has a yaw calibration error and therefore is generally well aligned with the inflow wind direction in the DD analysis period (i.e. each turbine exhibits a mean $\theta_{err}^V$ value of 0°). If this is true, then the wake of each turbine should roughly extend linearly downstream an angle consistent with the value of $\theta_{inf}^V$. To examine whether the wind turbine wakes were extending linearly downstream, the $T_L$, $T_T$, and $T_R$ wake orientation angles (i.e. $\theta_{wake}^V$) were compared to value of $\theta_{inf}^V$ in the figure below.

[Figure]

The left subplot demonstrates the values of $\theta_{wake}^V$ were consistently less than the volume-respective value of $\theta_{inf}^V$. Therefore, the wake centerlines were on average skewed to the left of their expected location based on the value of $\theta_{inf}^V$ and an assumed $\theta_{err}^V$ value of 0°. The right subplot demonstrates the difference between the $T_L$, $T_T$, and $T_R$ $\theta_{wake}^V$ values and the value of $\theta_{inf}^V$. On average, the $T_L$ $\theta_{wake}^V$ value was offset of $\theta_{inf}^V$ by -2.90°, the $T_T$ $\theta_{wake}^V$ value was offset of $\theta_{inf}^V$ by -3.34°, and the $T_R$ $\theta_{wake}^V$ value was offset of $\theta_{inf}^V$ by -2.17°. Negative values indicate wake skew to the left when looking downstream. Because the values of $\theta_{wake}^V$ were inconsistent with the volume-respective value of $\theta_{inf}^V$, and also because the $\theta_{skew}^V$ values demonstrated little sensitivity to variations in $\theta_{err}^V$, yaw calibration errors were not expected to have a significant impact on the presented results.

To provide the reader with more confidence that yaw calibration errors did not adversely impact the results, text was added to the manuscript.

Pg. 17 Lns. 2 through 4

"Furthermore, the values of $\theta_{\text{skew}}^V$ were inconsistent with $\theta_{\text{inf}}^V$, which along with the weak correlation between $\theta_{\text{skew}}^V$ and $\theta_{\text{err}}^V$ provides confidence that the unexpected mean sign of $\theta_{\text{skew}}^V$ was not simply a result of a turbine yaw calibration error."

**Pg. 11, ln. 15: 'However, nacelle-based measurements are inherently distorted...' Another factor to consider is that the flow distortion from the rotor can change as the control changes. Adding a pitch offset could cause the wind speed behind the rotor to change differently than with the original control, complicating the detection of changes in turbine operation as a function of wind speed.**

The referee indicates that experimental turbine control changes could impact the validity of the nacelle-transfer function (NTF). These turbine control changes could modify typical flow distortion levels behind the rotor, thereby reducing the effectiveness of the NTF used to convert the nacelle-mounted anemometer measurement to a rotor-effective inflow velocity. Therefore, experimental control changes could also impact detection of the control changes implemented when analyzed as a function of the NTF-based inflow velocity. As denoted prior, this demonstrates the need for more accurate turbine inflow estimates, in particular measurements that are independent of the turbine control system. Manuscript text was slightly modified to reflect the need for these types of measurements.

Pg. 12 Lns. 11 through 16:

"To improve controller assessment, future field campaigns should place precedence on turbine inflow measurements (wind speed and direction) independent of the turbine control system (i.e. non-SCADA data). Experiments using scanning-based measurements should use advanced analysis techniques, such as those established in Duncan et al. (2019) wherein space-to-time conversions were performed on the spatially distributed velocity fields, to provide a comprehensive characterization of the turbine inflow wind speed and direction on a second-by-second basis. Application of these methods was limited because of data availability issues."

**Pg. 12: ln. 10: 'Both of these factors might have contributed to the experimental control offsets not being fully realized.' Certainly for yaw control, a single 10-minute period might not be sufficient to observe meaningful yaw misalignment changes, given the slow dynamics of yaw controllers.**

Additional text was added to the manuscript to better convey this message.

Pg. 12 Lns. 25 through 27:

For example, the brief duration of the experimental periods may have been insufficient to realize/validate the desired control offsets (in particular the ability to observe significant yaw misalignment changes), and the ability to implement the desired control

changes might have been impacted by the ABL conditions present (e.g. region three inflow wind speeds).

**Section 4.1: I would suggest a revised wake tracking algorithm in light of improvements in the understanding of wake deflection physics. As discussed in papers such as the following, yaw misalignment can cause wakes to have a 'curled' shape due to the presence of counter rotating vortices. This means the peak velocity deficit could change with height and averaging across all heights in the rotor disk area is not necessarily the most relevant metric.**

- **Vollmer et al. Estimating wake deflection downstream of a wind turbine in different atmospheric stabilities: an LES study, Wind Energy Science, 2016**

- **Howland et al, Wake structure in actuator disk models of wind turbines in yaw under uniform inflow conditions, Journal of Renewable and Sustainable Energy, 2016.**

- **Fleming et al., A simulation study demonstrating the importance of large-scale trailing vortices in wake steering, Wind Energy Science, 2018.**

**A more meaningful lateral wake center estimate for wind plant control applications can be found using the method explained in Vollmer et al. 2016, where the cubed wind speed is averaged across a hypothetical rotor disk area centered at different lateral displacements. The displacement that results in the lowest value can be considered the wake center position.**

The referee contends the employed wake-tracking algorithm might not have adequately resolved lateral wake center location because of the impact yaw misalignment can have on wake structure. However, before addressing this concern, the authors would like to clarify the methods used in the wake-tracking algorithm. The referee states 'averaging across all heights in the rotor disk area is not necessarily the most relevant metric'; however, no averaging was used in the wake-tracking algorithm. Instead, at each constant-height plane within the wind turbine rotor sweep, the horizontal location of the wind speed minimum was determined and the median (not mean) horizontal location of these minima was used to discern the lateral wake center location.

Regardless, an analysis was performed to examine whether the operational yaw errors modified wake shape. Both numerical simulation and wind tunnel testing demonstrate rotor-sweep relative variations in wind turbine thrust caused by yaw error can promote a kidney-shaped (or curled) wake (e.g. Churchfield et al., 2016; Vollmer et al., 2016; Bartl et al., 2016). Therefore, wake structure changes induced via yaw error could potentially impact the wake-tracking algorithm. To briefly examine changes in wake shape due to yaw error, composite mean wake deficit cross-sections were developed for the $T_L$, $T_T$, and $T_R$ using data from all contributing DD volumes in the $+10°$ experimental period.

To determine the composite mean wake deficit cross-section, the vertical wake cross-sections used to identify the wake were first redefined at each distance downstream about the wake center location. Wind speed measurements within the redefined cross-sections were then converted to wake deficit values, defined as the percentage reduction in wind speed from the DD volume freestream wind speed profile. The DD volume freestream wind speed profile was developed by

determining the mean wind speed within the freestream analysis area at each constant-height plane within the wind turbine rotor sweep (i.e. the same as Fig. 5 but for wind speed instead of wind direction). Therefore, the individual wake deficit values (i.e. $v_{deficit}$) were defined by,

$$v_{deficit} = \frac{v_{wake}^{V,h} - v_{inf}^{V,h}}{v_{inf}^{V,h}} \cdot 100$$

where $v_{wake}^{V,h}$ denotes a wake cross-section wind speed measurement at height ($h$) and $v_{inf}^{V,h}$ denotes the height-respective DD volume freestream wind speed. The wake deficit cross-sections were then averaged at each distance downstream to determine the $T_L$, $T_T$, and $T_R$ composite mean wake deficit cross-sections for the +10° experimental period. Wake deficit contours at 10 %, 15 %, and 20 % were used to qualitatively examine wake shape. However, based on this contour analysis at 1 RD downstream (refer to the figure below), neither the wake of the $T_L$, $T_T$, nor $T_R$ exhibited a modified (i.e. kidney or curled) shape. The contours are roughly concentric, and thereby denote a near circular wake. The most notable difference between the composite mean wake of the $T_L$, $T_T$, and $T_R$ was the relative size of the contours (i.e. indicating differences in wake intensity).

[Figure]

**Reference Figure:** Composite mean wake deficit cross-sections (presented as a percentage reduction from the DD volume freestream wind speed profile) at 1 RD downstream for the (a) $T_L$, the (b) $T_T$, and (c) the $T_R$. Contours corresponding to a 10 %, 15 %, and 20 % wake deficit are overlaid in black and the wind turbine rotor sweep is overlaid in red. (d) The $T_L$, the $T_T$, and the $T_R$ 15 % wake deficit contour at 1 RD downstream.

While this analysis was limited to the +10° experimental period, additional analysis was performed using data from all DD volumes within the DD analysis period (i.e. 14:22:32 UTC – 15:31:57 UTC). At 1 RD, 2 RD, and 5 RD downstream, the mean horizontal (i.e. lateral) location of the wind speed minima (defined relative to the lateral location of the wake center [i.e. +/-]) was determined at each constant-height plant within the wind turbine rotor sweep for the $T_L$, $T_T$, and $T_R$. The mean lateral location of these wind speed minima is denoted in the figures below. Variability about the zero line denotes the average displacement of the height-respective wind speed minima from the wake center. Therefore, changes in wake structure due to yaw error should

be reflected in these mean profiles. Although there were some lateral displacements in the wind speed minima mean location at 1 RD downstream, no mean displacement exceeded 10 m. The observed displacements were not significant enough to indicate extensive wake shape deformation resulting yaw error. These results do not indicate yaw error does not modify wake shape, but suggest the operational yaw error values along with the turbine inflow wind speeds were not significant enough to elicit these changes. Therefore, the effectiveness of the wake-tracking algorithm is not expected to have been adversely impacted by any perceived changes in wake shape resulting from yaw error.

[Figure]

To ease referee concerns regarding the employed wake-tracking algorithm, the referee-recommended wake-tracking algorithm was also applied and analyses was re-ran. The lateral wake center location was determined at each distance downstream by identifying the lateral location that minimized the rotor-sweep disk averaged cubic wind speed. Lateral displacements ranging from -100 m to +100 m at 10-m intervals were examined at each distance downstream. Using this method (refer to the figure below), the $T_T$ wake at 14:33:20 UTC was located -10 m (i.e. to the left when looking downstream) of its expected location based on the inflow wind direction (i.e. $\theta_{inf}^V$).

[Figure]

Despite implementing the referee-recommended wake-tracking algorithm, the wake skew values were not significantly modified. With this wake-tracking algorithm applied, all three turbines still exhibited a negative mean wake deflection angle, indicating wake deflection to the left when looking downstream. This is opposite of expected based on the operational yaw error values and theory. Therefore, the choice of wake-tracking algorithm did not impact the take-home message of the wake skew analysis. In order to maintain consistency between the SD and DD wake-tracking

algorithms, the wake-tracking algorithm previously established by Hirth and Schroeder (2013) will be employed.

**Fig. 9: Is Fig. 9 (a) showing the distance from the centerline of the wake after corrected for the skew angle, or from the centerline in the mean wind direction. Please clarify what is being shown.**

The figure indicates the lateral distance from the wake centerline as defined by $\theta^V_{wake}$ (i.e. "after corrected for the skew angle"). This is denoted in the first manuscript submission on Pg. 15 Lns. 4 through 5:

> "Constrained linear least-squares regression (Gill et al., 1981) was used to determine the value of $\theta^V_{wake}$, wherein the wake centerline was required to emanate from the location of the wind turbine."

To improve clarity, both manuscript and figure text were modified.

Pg. 15 Ln. 13 and Pg. 16 Lns. 1 through 2:
> "The value of $\theta^V_{wake}$ minimized the error sum of squares; the error distribution was defined as the lateral distance between individual wake center locations and the wake centerline (i.e. $\theta^V_{wake}$) (e.g. Fig. 9a)."

Figure 9 Caption:
> **Figure 9.** (a) The lateral distance between the $T_T$ wake centerline (as defined by $\theta^V_{wake}$) and the wake center locations (i.e. the $T_T$ error distribution). (b) TTUKa DD hub-height wind speed (m s$^{-1}$) at 15:30:00 UTC overlaid by the $T_L$, $T_T$, and $T_R$ wake centerline and the wake center locations at 1-RD increments.

**Pg. 16, ln 6: '...indicating the observed wake deflection...was opposite of that expected.' How might wind veer impact the wake deflection during the experiment period? Could this be an explanation for the unexpected skew?**

Wind veer can induce rotor-sweep relative variations in yaw error. In the presence of wind veer, a rotor that is well aligned with the wind at hub height will exhibit increasing magnitudes of yaw error (albeit opposite signs) in the upper and lower halves of the wind turbine rotor sweep. Therefore, a turbine exhibiting minimal yaw error at hub height should produce a wake (as viewed from above) that widens with distance downstream. This perceived wake widening is due to the wake being deflected in one direction (left/right) in the lower half of the rotor sweep and in the other direction (right/left) in the upper half of the rotor sweep. Depending on the veer profile (i.e. veering/backing and intensity [° per m AGL]), wake deflection can either be promoted or inhibited relative to that expected based on the hub-height turbine inflow wind direction and the yaw orientation angle. For this reason, hub-height wind directions cannot comprehensively

quantify the rotor-relative variations in yaw error that contribute to wake deflection. To increase the robustness of the results, a rotor-sweep average wind direction (i.e. $\theta_{inf}^V$) was used to quantify yaw error. This was noted in the first manuscript submission on Pg. 9 Lns 16 through 18 and Pg. 10 Lns 1 through 5.

> "Due to wind plant and turbine measurement limitations, yaw error is traditionally defined relative to the hub-height wind direction measured by the nacelle wind vane. However, the nacelle wind vane is unable to account for differences in wind direction with height (i.e. wind veer). Therefore, yaw error defined by the hub-height wind direction will be unable to comprehensively quantify the rotor-sweep relative variations in the axial induction factor that cause wake deflection. Hence, yaw error $\left(\text{i.e. } \theta_{err}^V\right)$ was defined in each DD volume relative to the RSA average turbine inflow wind direction using
>
> $$\theta_{err}^V = \theta_{inf}^V - \theta_{yaw}^V$$
>
> where $\theta_{yaw}^V$ was the DD volume yaw angle (i.e. the mean yaw angle during the DD volume acquisition period). Positive magnitudes of $\theta_{err}^V$ denote counterclockwise rotor rotation relative to $\theta_{inf}^V$."

Prior to examining wind veer in the DD analysis period, it can be noted that the $T_L$, $T_T$, and $T_R$ wake center analysis performed at 1 RD, 2 RD, and 5 RD downstream (provided in a previous response) did not exhibit the opposing wake deflection signs in the upper- and lower-halves of the rotor sweep that would be expected if significant wind veer was present. To further demonstrate wind veer was not the cause of the unexpected wake skew angles, wind veer was quantified in each DD volume by analyzing the freestream wind direction profile (i.e. Fig. 5). Both the DD analysis period mean and one-sigma value of these freestream wind directions at each constant-height plane within the vertical depth of the wind turbine rotor sweep are provided in the figure below along with a time history of wind veer (defined as the freestream wind direction difference between 130 m and 30 m) in the DD analysis period. Within the DD analysis period, wind veer ranged from -4.16° per 100 m to +2.22° per 100 m about a mean value of -0.60° per 100 m. Over 80 % of the DD volumes exhibited wind veer magnitudes less than 2° per 100 m. Therefore, although wind veer can impact wake deflection, the wind veer present in the DD analysis period was not significant enough to adversely impact the results of the wake skew analysis.

[Figure]

**Pg. 18, ln 3: '9 degrees counterclockwise...of $\theta_{\text{inf}}^V$. Stating what $\theta_{\text{inf}}^V$ is would clear up any confusion about the sign convention.**

The value of $\theta_{\text{inf}}^V$ was defined in the first manuscript submission on Pg. 9 Lns. 13 through 16:

> "The freestream wind direction was defined at each DD constant-height plane within the vertical depth of the wind turbine rotor sweep (Fig. 5b), and the mean of these freestream wind direction measurements was used to determine the rotor sweep area (RSA) average turbine inflow wind direction $\left(\text{i.e. } \theta_{\text{inf}}^V\right)$."

However, because there is a gap in the usage of this term, manuscript text was modified to refer the reader back to Sect. 3.1.2 for the definition of $\theta_{\text{inf}}^V$.

Pg. 19 Lns. 4 through 5:
> "At 15:21:08 UTC, a $\theta_{\text{streak}}^V$ value of 162.72° was determined, which was 9° counterclockwise (i.e. to the left when looking downstream) of $\theta_{\text{inf}}^V$ (defined in Sect. 3.1.2)."

**Section 4.2.1: The potential impact of streak orientation on wake skew is an interesting idea. However, a deeper discussion of how this might cause the skewing of the wake would be appreciated.**

In the first manuscript submission (Pg. 17, Lns. 4 through 5), it was denoted that previous research demonstrates boundary layer streak orientation can slightly deviate (i.e. clockwise or counterclockwise) from the ABL wind direction. While Lorsolo et al. (2005) is listed as the primary reference for this research, both Morrison et al. (2005) and Foster (2005) have also identified streak orientation angles that deviate from the ABL wind direction. Despite this empirical research, it is unknown why boundary layer streak orientation differs from the ABL wind direction. Rather than presenting and validating hypotheses for the formation of boundary layer streaks, previous research has commonly focused on simply quantifying the spatial and temporal structure of these boundary layer heterogeneities. However, to provide the reader with more background on this research area, the references for Morrison et al. (2005) and Foster (2005) are now provided in the manuscript.

Because the physical mechanisms that govern boundary layer streak orientation are relatively unknown, it is difficult to hypothesize how these streaks directly govern downstream wake progression. However, because the dimensions of boundary layer streaks scale well with wind turbine wakes, it can be hypothesized that the same boundary layer forcings causing streak orientation to deviate from the ABL wind direction might also promote similar deviations in wake orientation. To convey this to the reader, manuscript text was modified.

Pg. 18 Lns. 6 through 7:
> "Although it is not fully understood what causes streak orientation to differ from the ABL wind direction, it can be hypothesized that the same boundary layer forcings also promote downstream wake deviation from the ABL wind direction."

Comprehensively quantifying how boundary layer streaks modulate downstream wake progression is outside the scope of this manuscript. However, the authors hope that establishing this preliminary connection will stimulate future research that examines how boundary layer heterogeneities impact the effectiveness of various wind plant control methods.

**Fig. 13: Consider showing the joint probability density of streak skew angle and wake skew angle. This would support your idea of a correlation between the two better.**

Thank you for your input. A joint probability density of $\theta^V_{\text{S-skew}}$ versus $\theta^V_{\text{skew}}$ is now provided in the manuscript (i.e. Fig. 14). Corresponding to Fig. 14, the following text was incorporated into the manuscript.

Pg. 19 Lns. 11 through 15:
> "Similarities between the $\theta^V_{\text{S-skew}}$ and the $T_L$, $T_T$, and $T_R$ $\theta^V_{\text{skew}}$ distributions are further demonstrated by examining their joint probability density. In Fig. 14 the $T_L$, $T_T$, and $T_R$ $\theta^V_{\text{skew}}$ distributions were combined and compared to the $\theta^V_{\text{S-skew}}$ distribution; darker shades of blue indicate increasing probability of the respective $\theta^V_{\text{S-skew}}$ and $\theta^V_{\text{skew}}$ values. Despite positive values of $\theta^V_{\text{S-skew}}$, a large percentage of the skew angles (both $\theta^V_{\text{S-skew}}$ and $\theta^V_{\text{skew}}$)

exhibited counterclockwise rotation relative to $\theta_{inf}^V$ (indicating downstream deflection to the left)."

[Figure]

**Figure 14.** The joint probability density between the $\theta_{S\text{-skew}}^V$ and $\theta_{skew}^V$ distributions. The $T_L$, $T_T$, and $T_R$ $\theta_{skew}^V$ distributions were combined to produce this heatmap, wherein darker shades of blue indicate an increasing probability of occurrence for the respective $\theta_{S\text{-skew}}^V$ and $\theta_{skew}^V$ values.

**Section 5.1: SD WTA: For a section title, consider spelling out the acronym.**

The acronym WTA is now spelled out in the section title.

**Section 5: In addition to wake length and wake meandering, what differences have you observed in the relative magnitude of the velocity deficits for stable vs unstable conditions? This would be a valuable addition to the paper.**

The authors agree that this would be a valuable addition to the paper. However, analysis and comparison of the convective and stable ABL wake deficit profiles was hindered by several factors. First, there were significant differences in the turbine inflow wind speed between the convective and stable ABLs (reflected in Fig. 15 [now Fig. 16] using 10-min average data). Although 10-min average wind speeds were provided in the manuscript, high-temporal resolution wind speed data spanning the SD analysis period is provided in the figure below to demonstrate these wind speed differences. The red and blue shaded regions denote the convective and stable portions of the SD acquisition period, respectively. The amount of momentum extracted from the inflow, and therefore the wake deficits immediately downstream, inversely vary as a function of the turbine inflow wind speed. Therefore, it would be difficult to exclusively attribute difference

in the convective and stable wake deficit profiles to changes in ABL stability, rather than simply differences in the turbine inflow wind speed.

[Figure]

A second confounding factor to wake deficit analysis was the variation in measurement height with range (approximately 17.5 m per km moving away from the radar [noted in the manuscript on Pg. 5 Lns. 10 through 11]). This impacted the potential for wake deficit analysis as follows:

1) Wind shear varies between the convective and stable ABLs. In the stable ABL, wind shear is typically larger than in the convective ABL. Therefore, the turbine inflow wind speeds will be inherently larger than the wake velocities because of their relative measurement range and height. A shear-correction would be required to resolve relevant wake deficit profiles.

2) Because measurement height varies with range, different portions of the wake are examined at incremental distances downstream.

For these reasons, it is difficult to know how relevant the extracted wake deficit profiles would be for research purposes, or how well they would correlate with wake deficits found in previous research. Wake length was instead examined as a proxy to discern the downstream extent of the wake effect in both the convective and stable ABLs.

To more clearly convey differences in wind speed (and atmospheric turbulence) between the convective and stable ABLs, the convective and stable portions of the SD analysis period are now shaded in Fig. 15 (now Fig. 16). The revised figure was copied below for reference.

James B Duncan Jr
https://doi.org/10.5194/wes-2019-78-RC1

[Figure]

**Pg. 26: ln. 14: 'access to the controller design so any factors inhibiting proper implementation of the turbine control offsets can be identified.' I agree that access to the controller improves the assessment of wind plant control strategies, and is always desirable, but I think meaningful control assessments can be done without direct access. For example, adding a pitch offset in region 2 (where pitch is typically fixed at 'fine pitch') could be achieved without needing to understand the controller dynamics. Furthermore, to implement a yaw misalignment, the yaw controller setpoint could be changed from zero to the desired offset, but full understanding of the controller dynamics is not necessary, and in many cases would be asking too much given the proprietary nature of wind turbine control systems.**

Meaningful wind plant control experiments could be performed without full access to wind turbine controller design. However, a comprehensive analysis of the controller's ability to implement the desired control strategies would likely require access to its control structure and design. Without this information, it would be difficult to discern why the turbine was able/unable to implement the desired control changes and it would also be challenging to determine how these control changes could be more efficiently employed. However, manuscript text was modified to soften reader interpretation.

Pg. 28 Lns. 20 through 21:
    "...(2) access to relevant controller information (e.g. controller design) so any factors inhibiting proper implementation of the turbine control offsets can be identified."

---

## Author Comment (AC2) · 28 Jan 2020

James B Duncan Jr
https://doi.org/10.5194/wes-2019-78-RC2

**General Response**

**Thank you for this paper. It is very useful to receive experimental results, and it is always a major undertaking to gather such data. In general the paper is well written with good and useful figures.**

**My major criticism/suggestion is that the results presented in sections 3/4 are of too little data, and with too many "black box" issues to be used to draw conclusions from. Section 5 on the other hand provides results which are useful, line up with physical interpretation, and show statistical significance. I would therefore propose to condense (or remove?) sections 3/4, and perhaps expand a bit on section 5.**

Thank you for your comments and the time taken to provide this review. The authors are glad you appreciate the experimental data and are understanding of the difficulties inherent to data collection.

The main criticism of the referee is that the results presented in Sections 3 (Wind Plant Control Experimental Setup, Controller Assessment, and Controller Assessment Challenges) and 4 (Measuring Wind Turbine Wake Response to First-Order Turbine Control Changes) are of too little data and with too many "black box" issues to draw conclusions from. The authors agree there are innate limitations to the dataset. However, this is why the initial focus of the manuscript was modified as analyses progressed to, in part, exploring the difficulties associated with performing a wind plant control experiment at full scale.

At the time of the experiment (14 December 2014), experimental validation of wind plant control at full scale was rare. Similarly, there was a dearth of observational data (especially compared to numerical simulation) characterizing in three-dimensions wind turbine wake structure and variability. Therefore, in collaboration with an industry partner, the preliminary objectives of the experiment were to (1) examine three-dimensional wind turbine wake response to changes in wind turbine yaw and blade pitch and (2) examine how these changes impact the net power production of individual turbines in the wind plant (i.e. quantifying the effectiveness of the wind plant control strategy). However, experimental logistics (e.g. experimental control limitations imposed by the wind plant operator) ultimately inhibited the successful execution of these experimental objectives. As noted in our response to referee one, the wind plant control experiment would have been ideally performed in an environment more conducive to the effectiveness of the implemented control strategies and for longer experimental durations. Still, the authors are confident that there is value to disseminating the results of this experimental campaign, albeit in a suboptimal environment and subject to innate limitations.

The results of this study lend insight into the complexities associated with performing a wind plant control experiment at full scale, and therefore, should be used to inform future field campaigns. Section 3 provides methods that can be referenced in future studies, but furthermore, this section highlights the importance of several key data sources and how not having access to these data can severely impact both analyses potential and the ultimate success of the field campaign. Section 4 additionally establishes that strategic yaw error in some ABL environments might not be sufficient to ensure effective wake steering. The authors were not intending based off this analysis to provide

James B Duncan Jr

https://doi.org/10.5194/wes-2019-78-RC2

a statistical characterization of the wake. Rather, the authors are optimistic these results will promote further research examining how certain ABL conditions and transient ABL heterogeneities might impact both the effectiveness and potential benefit of various wind plant control strategies.

The results of Section 4 also provide a smooth transition to Section 5 (ABL stability driven wake changes). The referee suggests that the analysis in Section 5 should be expanded. Although this is desirable, SD wake analysis was also limited. For example, referee one advised that velocity deficits between the convective and stable ABLs be examined. However, analysis and comparison of the convective and stable ABL wake deficit profiles were hindered by several factors. First, there were significant differences in the turbine inflow wind speed between the convective and stable ABLs (reflected in Fig. 15 [now Fig. 16] using 10-min average data). Although 10-min average wind speeds were provided in the manuscript, high-temporal resolution wind speed data spanning the SD analysis period is provided in the figure below to demonstrate these differences in wind speed. The red and blue shaded regions denote the convective and stable portions of the SD acquisition period, respectively. The amount of momentum extracted from the inflow, and therefore the wake deficits immediately downstream, inversely vary as a function of the turbine inflow wind speed. Therefore, it would be difficult to exclusively attribute difference in the convective and stable wake deficit profiles to changes in ABL stability, rather than simply differences in the turbine inflow wind speed.

[Figure]

A second confounding factor to wake deficit analysis was the variation in measurement height with range (approximately 17.5 m per km moving away from the radar [noted in the manuscript on Pg. 5 Lns. 10 through 11]). This impacted the potential for wake deficit analysis as follows:

1) Wind shear varies between the convective and stable ABLs. In the stable ABL, wind shear is typically larger than in the convective ABL. Therefore, the turbine inflow wind speeds will be inherently larger than the wake velocities because of their relative measurement range and height. A shear-correction would be required to resolve relevant wake deficit profiles.

2) Because measurement height varies with range, different portions of the wake are being examined at incremental distances downstream.

For these reasons, it is difficult to know how relevant the extracted wake deficit profiles would be for research purposes, or how well they would correlate with wake deficits found in previous research. Wake length was instead examined as a proxy to discern the downstream extent of the wake effect in both the convective and stable ABLs.

Despite the inherent limitations of the datasets presented, the authors are confident they provide a valuable contribution to the scientific community by highlighting links between the ABL and wake structure and variability and how this might impact the effectiveness of wind plant control. Furthermore, exploring the difficulties associated with performing a wind plant control experiment at full scale should help improve future experimental design.

**Specific Comments**

**P 2 "operate below their peak capacity to decrease wake effect..." this describes well static induction control, but less well wake steering and dynamic induction control**

Thank you for your comment. The manuscript text was slightly modified to improve reader comprehension. The edited text is provided below.

Pg. 2 Lns. 12 through 14:
    "When wind plant control is employed, some turbines in the wind plant will modify their control settings (sometimes operating below their peak capacity) to decrease the wake effect, thereby increasing the plant-wide available kinetic energy (De-Prada-Gil et al., 2015)."

**Fig 1: It's explained later, but the legend is unclear in meaning, perhaps explain more in caption**

Thank you for noting that the figure was not adequately described, the other reviewer shared a similar position. As you mention, the figure was initially detailed later on in the manuscript in Section 3 (Pg. 5 Lns. 11 through 16 and Pg. 6 Lns. 1 through 2). However, upon reconsideration, the authors recognize that this text is disconnected from the description of the 14 December 2014 deployment (i.e. Section 2.1) and Fig. 1. Therefore, this text was moved to Section 2 and was slightly modified to more comprehensively describe Fig. 1. The modified text is provided below.

Pg. 4 Lns. 14 through 23:

"Located in the DD domain of the 14 December 2014 deployment were 20 wind turbines distributed across two turbine rows. The wind turbines were characterized by a hub height of 80 m and a rotor diameter (RD) of 101 m. Supervisory control and data acquisition (SCADA) information detailing the turbine inflow wind speed (subject to the nacelle transfer function [NTF]), turbine yaw orientation, and blade pitch angle were provided at a one-hertz sampling frequency from 14:00:00 UTC to 16:59:45 UTC for seven of the wind turbines (denoted by the non-black circles in Fig. 1). Three of the seven wind turbines were located in the lead row of the wind plant (denoted by the blue, red, and purple circles in Fig. 1), while the remaining four were located in the trailing row (denoted by the white circles in Fig. 1). The three lead-row wind turbines were separated by an average distance of 1512.2 m (~15 RD) from the trailing turbine row and were laterally separated from each other by an average distance of 321.1 m (~3 RD). The wake of these three lead-row wind turbines (referred to as the $T_L$, $T_T$, and $T_R$) were analyzed to examine the effectiveness of the implemented wake-mitigating control strategies."

Furthermore, both the Fig. 1 caption and legend were modified to improve reader comprehension. The updated figure and revised caption are provided below.

[Figure]

**Figure 1.** (a) Schematic of the TTUKa DD radar deployment on 14 December 2014 including radar deployment locations (red and blue squares), the radar sectors scanned (defined by the red and blue outlined regions), the DD domain (shaded black region), the location of the individual wind turbines (colored circles [the meaning of the different turbine colours are defined in Sect 2.1]), the mean wind direction (black arrow), and the underlying mean sea level elevation (m). (b) TTUKa DD hub-height wind speed (m s⁻¹) at 14:59:29 UTC overlaid by the wind turbine locations.

**P6: RD?**
RD is an acronym for rotor diameter and was established in the initial manuscript submission on Pg. 5 Lns. 12.

> "The wind turbines were characterized by a hub height of 80 m and a rotor diameter (RD) of 101 m."

This manuscript text was shifted up as part of the changes made to Fig. 1. The authors can confirm that the acronym RD was pre-established prior to its use in the manuscript.

**Bottom p9: Could alternatively define wind direction as the average yaw position of non-changed turbines?**

Thank you for your input. The referee suggests that wind direction could be defined as the average yaw position of non-changed turbines as opposed to using the freestream wind direction (i.e. $\theta_{inf}^V$). However, the authors believe that an average wind direction based on DD measurements in the upstream 1.45 km by 1.8 km freestream analysis area is more appropriate for analyses for several reasons. One reason for opting to use $\theta_{inf}^V$ instead of the average yaw position of non-changed turbines is the timescales of interest in the manuscript. Wake deflection is examined on a volume-by-volume basis, or approximately at 60-s time intervals consistent with the DD volume acquisition period (i.e. 60.4 s). Therefore, to ensure the robustness of the results, the wind direction estimate used to define wake deflection should be determined at similar time intervals (i.e. $\leq 60.4$ s). While the average yaw position of the non-changed turbines could be determined at these timescales, they are not expected to provide a more accurate estimate of the turbine inflow wind direction because of the construct of the wind turbine yaw controller. Refer to Pg. 9 Lns. 6 through 8 of the initial manuscript submission:

> "Unlike the construct of the blade pitch controller, a wind turbine will not actively yaw on a second-by-second basis to ensure optimal rotor alignment. A wind turbine will typically only yaw when the yaw error has exceeded some threshold (e.g. $\pm 10°$) for an extended period of time (e.g. 10 min)."

Therefore, as long as the yaw error does not exceed this pre-established threshold, the wind turbine will not modify its yaw to be better aligned with the inflow. While at larger timescales the wind turbine might be expected to exhibit little to no yaw error, this is not true for the minute timescales used in the presented wake deflection analysis (as demonstrated in the figure below). Provided in the top subplot is a time history of the $T_L$, $T_T$, and $T_R$ yaw orientation angles and provided in the bottom subplot is a time history of the $T_L$, $T_T$, and $T_R$ yaw error angles (i.e. $\theta_{err}^V$) as defined relative to $\theta_{inf}^V$. The vertical dashed lines denote the temporal bounds of the $+10°$ experimental period (i.e. 15:22:00 UTC – 15:31:59 UTC). Despite the $T_L$, $T_T$, and $T_R$ yaw controller remaining unmodified between 14:22:32 UTC and 15:22:00 UTC, each turbine exhibited significantly different yaw orientation angles. During this period wherein no turbine yaw changes were implemented, the $T_L$ exhibited a mean yaw orientation angle of 160.6°, the $T_T$ exhibited a mean yaw orientation angle of 156.4°, and the $T_R$ exhibited a mean yaw orientation angle of 168.5°.

[Figure]

Considering each turbine exhibits yaw error (and a wide range of yaw error values at that) in this period, the authors do not believe using the average yaw position of 'non-changed' turbines is more appropriate than using $\theta_{inf}^{V}$ to define the yaw deflection angles.

**Fig 8: Believe these wake directions are convex in the wrong direction, the wake deflection appears to be accelerating as heading downstream, whereas expectation would be recovery to main direction (cf fig 1 in Jiménez, Ángel, Antonio Crespo, and Emilio Migoya. "Application of a LES Technique to Characterize the Wake Deflection of a Wind Turbine in Yaw." Wind Energy, 2010. https://doi.org/10.1002/we.) this might also impact analysis in fig 10.**

The referee notes that Fig. 8 indicates wake deflection increases with distance downstream. This was not the intended interpretation of the figure, instead the figure was meant to denote the general wake deflection directions (i.e. right/left) when certain yaw error angles were present (i.e. positive/negative). This is why no units were placed on either the X or Y axis. However, the authors do recognize that the reader could easily misinterpret the figure. Therefore, the figure was modified to more accurately convey the intended message. The revised figure and caption are provided below.

James B Duncan Jr
https://doi.org/10.5194/wes-2019-78-RC2

[Figure]

**Figure 8.** Wake deflection directions theoretically induced by (a) counterclockwise (i.e. $\theta_{\mathrm{err}} > 0$) and (b) clockwise (i.e. $\theta_{\mathrm{err}} < 0$) yaw rotation relative to a fixed turbine inflow wind direction.

The referee also correctly states that even though yaw error can promote wake deflection in a particular direction, this deflection angle will not continue infinitesimally downstream. At some downstream distance (as noted in Fig. 1 of Jiménez et al. [2010]), downstream wake progression will become more consistent with the governing wind direction. Although this is true, the authors are confident that this downstream wake behavior did not significantly impact the results presented in Fig. 10.

First, let the authors clarify that wake deflection was defined relative to $\theta_{\mathrm{inf}}^{V}$ and not the pseudo-expected lines curves initially depicted in Fig. 8. Secondly, the aptitude of downstream wake progression to return to values consistent with the governing wind direction should slightly reduce the absolute value of the derived wake deflection angles (i.e. $\theta_{\mathrm{skew}}^{V}$). Therefore, had this downstream wake progression impacted results, it would also indicate that wake deflection in the near-wake region was likely more anomalous (i.e. further to the left) than that indicated by the mean $T_{L}$, $T_{T}$, and $T_{R}$ values of $\theta_{\mathrm{skew}}^{V}$. Therefore, a return of the downstream wake progression to directions consistent with the mean flow would not have impacted the take-home message of the wake deflection analysis—the mean $T_{L}$, $T_{T}$, and $T_{R}$ $\theta_{\mathrm{skew}}^{V}$ values were opposite of that expected, indicating that some ABL characteristics (e.g. boundary layer streak orientation) might be instead governing wake orientation and downstream wake progression.

James B Duncan Jr
https://doi.org/10.5194/wes-2019-78-RC2

Regardless, the authors did consider the potential impact of downstream wake progression returning to directions consistent with the mean flow when performing analysis. For example, assume wake center locations outwards of only 6.5 RD downstream were considered (i.e. instead of the 13 RD used in the manuscript). Using only these wake centers to derive $\theta_{skew}^V$ results in the figure below (i.e. the same as Fig. 10 but using a different wake center distribution) and a mean $T_L$ $\theta_{skew}^V$ value of -2.15°, a mean $T_T$ $\theta_{skew}^V$ value of -2.06°, and a mean $T_R$ $\theta_{skew}^V$ value of -1.27°. Therefore, these results indicate that for the downstream distances examined (i.e. 13 RD) any tendency for downstream wake progression to return to values consistent with the mean flow did not significantly impact the presented results.

[Figure]

**P 19 "simply implementing yaw error might not be enough to ensure effective wake steering" not clear this result can be drawn from these results**

Demonstrated in the figure below, both the $T_L$, $T_T$, and $T_R$ exhibited yaw error in the DD analysis period. The DD analysis period mean $\theta_{err}^V$ value was positive for each turbine (refer to Fig. 6 of the manuscript), yet the mean $\theta_{skew}^V$ values were negative for each turbine in the DD analysis period. These negative $\theta_{skew}^V$ values are opposite of expected based on theory. Therefore, positive values of $\theta_{err}^V$ were not sufficient given the underlying ABL conditions to elicit the expected wake deflection to the right when looking downstream.

[Figure]

However, the authors note that the manuscript text does not adequately denote why simply implementing yaw error might not be sufficient to ensure effective wake steering. It is trivial given previous research to note that yaw-induced wake deflection is possible. However, it is not fully understood how the effectiveness of yaw-based wake steering varies given the prevailing ABL conditions. Therefore, the manuscript text was modified to indicate that in some ABLs (e.g. a convective ABL or in the presence of certain ABL heterogeneities such as ABL streaks) simply implementing yaw error might be sufficient to ensure effective wake steering.

Pg. 20 Lns. 10 through 13:
> "Regardless, these results are important because they suggest that in certain ABLs simply implementing yaw error might not be sufficient to ensure effective wake steering; rather, an integrated knowledge of ABL heterogeneities and their characteristics is needed (e.g. interaction with these transients might amplify or inhibit wake deflection)."

**Fig 17 and Fig 20: Great figures and really interesting!! Text analysis also interesting**

Thank you for your comment. The authors are glad that you found the analyses to be interesting.

---

## Author Comment (AC3) · 28 Jan 2020

[revised manuscript text omitted]